

# The influence of antecedent conditions on flood risk in sub-Saharan Africa

Konstantinos Bischiniotis[1], Bart van den Hurk[1,2], Brenden Jongman[1,3], Erin Coughlan de Perez[1,4,5], Ted Veldkamp[1], Jeroen Aerts[1]

(1) Institute for Environmental Studies, Vrije Universiteit Amsterdam, 1081 HV Amsterdam, The Netherlands

(2) Royal Netherlands Meteorological Institute (KNMI), De Bilt, 3731 GA, the Netherlands

(3) Global Facility for Disaster Reduction and Recovery (GFDRR), World Bank, Washington DC, USA

(4) International Research Institute for Climate and Society, Columbia University, Palisades, NY 10964, USA,

(5) Red Cross Red Crescent Climate Centre, The Hague, 2521 CV, the Netherlands

*Correspondence to*: Konstantinos Bischiniotis (kbischiniotis@gmail.com)

**Abstract** Most flood early warning systems have predominantly focused on forecasting floods with lead times of hours or days. However, physical processes during longer -seasonal- time scales can also contribute to flood generation. In this study, the hydro-meteorological pre-conditions of 501 historical damaging flood events over the period 1980 to 2010 in sub-Saharan Africa are analysed. These are separated into a) a short-term 'weather

scale' period (0-7 days) and b) a long-term 'seasonal-scale' period (up to 6 months) before the flood event. Total 7-day precipitation is used to evaluate weather-scale conditions, while seasonal-scale conditions are drawn from the Standardized Precipitation Evapotranspiration Index (SPEI). Although the latter has been used for drought detection, because of its characteristics can also become a wetness monitoring tool. Results indicate that, although heavy 7-day lead precipitation is connected with most reported floods (72%), more than half exhibited

wet antecedent conditions during the 6 preceding months. In case of extremely wet weather and seasonal scale conditions (SPEI > 2) the probability of flood is close to 50%. The combined analysis of the two periods revealed that seasonal-scale information should not be neglected, and SPEI information could be a useful -additional- input to the weather-scale flood forecasts to improve flood preparedness.

## 1 Introduction

In recent decades, weather-related disasters have accounted for about 90% of all natural disasters (UNISDR, 2015a). There is an upward trend in disaster loss, which is driven by global climate change and the increasing concentration of populations and economic assets in flood-prone areas (Bouwer et al. 2007; Prenger-Berninghoff et al. 2014). Flooding affects millions of people across the globe, and between 1980 and 2012 the average annual reported losses and fatalities due to floods exceeded $23 billion and 5,900 people, respectively (EM-DAT, 2012; Jongman et al., 2015).


Flood risk management has traditionally focused on long-term flood protection techniques such as levees and dams (Kellet and Caravani, 2013). In recent years, however, various new combinations of flood risk strategies have been developed, ranging from technical flood protection measures to financial compensation mechanisms such as insurance, as well as nature-based solutions (Aerts et al., 2014). Lower-income countries are often not able to afford and implement preventive measures,

mainly due to the high investment costs (e.g. Douben, 2006). Consequently, they are more reliant on post-disaster response and preparedness activities, often assisted by international donors and humanitarian organizations.





Preparedness activities and flood forecasting have received increasing attention and have led to new science-based early warning systems (Kellet and Caravani, 2013; Coughlan de Perez et al., 2014). Such forecasting systems, with a typical lead time of some hours or days, have reduced flood impacts not only in developed countries (Rogers and Tsirkunov, 2010), but also in several lower-income countries (Golnaraghi, 2010; Webster, 2013). Weather forecasts have become the basis of such

systems (Alfieri et al., 2012). Research has shown that the devastating 2010 Pakistan floods could have been predicted 6-8 days in advance if quantitative precipitation forecasts had been available, providing sufficient time for reaction (Webster et al., 2011). On longer time scales, seasonal forecasts can be helpful by signalling an increased likelihood of flooding, enabling inhabitants and relief organizations to prepare for action against floods at an early stage (Coughlan De Perez et al., 2015). However, forecast skill is inversely proportional to lead time (Molteni et al., 2011), and consequently the likelihood of taking

action in vain increases along with the warning system lead time.

One of the regions most vulnerable to flooding is sub-Saharan Africa, which contains a variety of climate zones (Peel et al. 2007). Six of the ten most flood-prone countries are situated in this region (UNISDR, 2015a). These countries either lack hard protective infrastructure against flooding or have a design protection level much lower than that in place in high-income

countries (Scussolini et al. 2016). Hence, early warning and timely preparation play an important role in risk reduction. Seasonal forecast information has been used successfully in West Africa by the International Federation of Red Cross and Red Crescent Societies (IFRC), reducing the risk from extreme rainfall (Tall et al., 2012; Braman et al., 2013).

These advances have led to novel forecast-based financing systems that automatically trigger humanitarian action based on

standard operating procedures when threshold forecasts are issued (Coughlan De Perez et al., 2015). These systems, however, still face uncertainties. One of the main issues is that research on seasonal flood forecasting has predominantly focused on precipitation as the main predictive factor, but other factors also play a role (e.g. Coughlan et al., 2017). For example, research has indicated the significance of the anomalous positive terrestrial water storage, including soil moisture, in the Missouri floods of 2011 (Reager et al., 2014). Other studies have shown the large influence of antecedent moisture, rather

than high rainfall, on the June 2013 floods in Germany (Schröter et al., 2015). Evapotranspiration and soil saturation are other factors considered important in flood forecasting (Sivapalan et al., 2005; Merz et al., 2006; Parajka et al., 2010; Fundel and Zappa, 2011). These physical factors may influence the length of the flood build-up period, which can range from a few days to several months before an event (Nied et al., 2014).

This study assesses the role of the antecedent conditions on short to long time scales in flood generation from 1980 to 2010 in sub-Saharan Africa and discusses whether seasonal-scale forecasts could be complementary to the weather-scale flood forecasting. More specifically, we analyse the correlation between reported floods and meteorological variables, both on a weather time scale (1-7 days before each flood event), and on a seasonal scale (up to 6 months before each flood event). Antecedent wet conditions were drawn from the Standardized Precipitation Evapotranspiration Index (SPEI) and additional

precipitation data. The SPEI is based on precipitation and temperature and records the wetness/dryness of an area. The index has been applied in studies focusing on seasonal forecasting of droughts, but not of floods (Mossad and Alazba, 2015; Xiao et al., 2016). The findings of this study contribute to the emerging literature on this topic (Goddard et al., 2014; White et al., 2015) and may be of use to humanitarian organizations and decision-makers for preventive flood risk management planning.

The remainder of this paper is structured as follows. Section 2 outlines the methodological framework and the data used in the analysis, followed in Section 3 by the results. Section 4 discusses the findings and the limitations of the study, including suggestions for further research. Section 5 provides a brief conclusion.

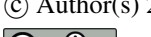



## 2 Methodology

Figure 1 shows the different steps in the approach taken by this study. The analysis is based on reported damaging flood events provided by Munich RE (2014). We assessed the antecedent weather and climate conditions in the locations of reported floods using two indicators: (a) the long-term ('seasonal-scale') wetness reflected in the SPEI for the preceding 1, 3
and 6 months, and (b) the short-term ('weather-scale') cumulative rainfall over the 7 days preceding the event.

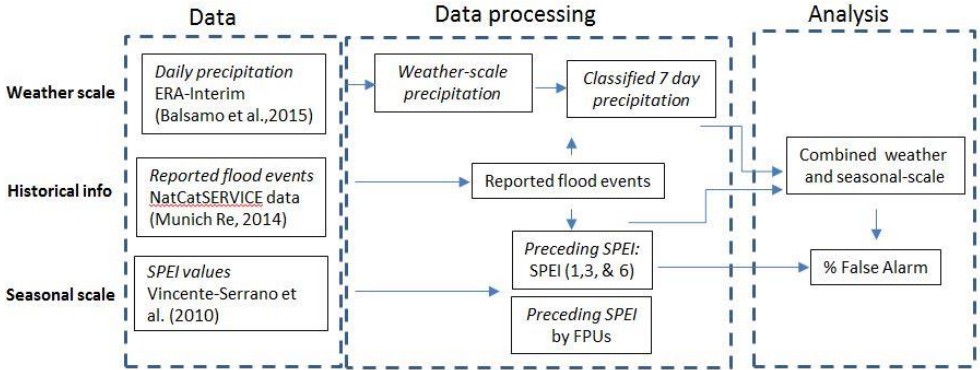

**Figure 1: Schematic overview of the approach followed in this study.**

### 2.1 Data

*Study area and reported floods*

We used the Munich Re NatCatSERVICE disaster database (Munich Re, 2014) to identify the reported flood events in sub-Saharan Africa. This database shows different characteristics for each flood event, such as flood duration, number of fatalities, economic losses, and the onset and end date of the event. Each event is georeferenced using a pair of coordinates,
which we used to place it on a 0.5° x 0.5° grid cell.

*Daily precipitation*

Daily precipitation was derived from the European Centre for Medium-Range Weather Forecasts (ECMWF) global
reanalysis of land-surface parameters, ERA-Interim/Land, over the period 1980-2010 (Balsamo et al., 2015) (available online at http://apps.ecmwf.int/datasets/). The data series were produced by applying the HTESSEL land surface module and the ERA-Interim precipitation dataset, and further processed using bias correction to match monthly accumulated precipitation totals (Dee et al., 2011). The gridded daily time series were deliberately extracted at 2.5° x 2.5° horizontal resolution, corresponding to the flooded areas, which are 64,000 km$^2$ on average (Douben 2006). This resolution was also used by
Coughlan et al. (2006) and is chosen here in order to capture the precipitation of the neighbouring areas of the flooded cell that could have contributed to flood generation during the 7-day period. Another reason for using this simplification is that the possible errors and omissions in the reported coordinates of an event make it difficult to determine the exact delineation of the upstream area.

*Standardized Precipitation Evapotranspiration Index (SPEI)*

The SPEI, developed by Vicente-Serrano et al. (2010) was used to evaluate the antecedent soil conditions before the reported flood events. Using a long time-series of at least 50 years, the SPEI compares monthly net precipitation totals (NP,



precipitation – potential evapotranspiration) with their long-term means over different time scales (1, 3, 6 or 12 months). An x-month SPEI provides a comparison over the same x-month period for all years in the historical record. Shorter accumulation periods (1 month) represent surface soil water content, while longer ones (3, 6, 12 months) indicate the subsurface state (e.g. soil moisture, groundwater discharge) (Du et al., 2013). Unlike the Standardized Precipitation Index

(SPI), the SPEI takes potential evapotranspiration into account, which can consume a large portion of total rainfall (Abramopoulos et al., 1988). Precipitation and evapotranspiration together largely determine soil moisture variability, and thus indirectly affect the flood build-up period through links between soil moisture, river discharge, and groundwater storage (Vicente-Serrano et al., 2010). Although some studies have successfully applied SPI as a flood indicator (Seiler et al. 2002; Guerreiro et al. 2008), SPEI has, to our knowledge, not yet been applied.

In this study SPEI values were derived at a 0.5° x 0.5° spatial resolution (available online at http://sac.csic.es/spei/index.html), focusing on the local conditions of the flooded area. Mean monthly temperature from the NOAA GHCN_CAMS gridded dataset (Fan and van den Dool, 2008) and mean monthly precipitation from the Global Precipitation Climatology Centre (GPCC) (Schneider et al., 2015) beginning in 1950 were used to estimate the monthly

potential evapotranspiration (PET), as in Thornthwaite (1948). Such a long-time dataset was not available through the ERA-Interim. A statistical procedure was applied to fit the accumulated records to a log-logistic distribution with a mean of 0 and a standard deviation of 1. The SPEI values are given for the end of each calendar month (see Vicente-Serrano et al., 2010, for more detail on the processing of the SPEI index). Hence, positive and negative SPEI values indicate wet and dry periods respectively (Table 1).

| SPEI class | Class description |
|---|---|
| ≤-2 | Extremely dry |
| -2 : -1.5 | Severely dry |
| -1.5 : -1 | Moderately dry |
| -1 : -0.5 | Mild drought |
| -0.5 : 0 | Near normal dry |
| 0 : 0.5 | Near normal wet |
| 0.5 : 1 | Mild wet |
| 1 : 1.5 | Moderately wet |
| 1.5 : 2 | Severely wet |
| >2 | Extremely wet |

**Table 1: Classification of SPEI values (Edossa et al., 2014).**

### 2.2 Data

*Temporal scale*

Figure 2 shows an example of discharge in the different phases of a flood event. The start date of each flood was derived from the NatCatSERVICE flood dataset (Munich Re, 2014). The period before the event is the 'flood build-up' period, during which we assumed that the physical processes that led to flooding took place (Nied et al., 2014). The period after the flood's end date is called 'attenuation period'.


The build-up period was divided into two parts: a preconditioning period at the seasonal scale (6 months lead time), and a flood triggering episode at the weather-scale period (7 days lead time). This made it possible to distinguish between the





antecedent conditions that may have led to an increased likelihood of flooding from the intense rainfall prior to the event. Although other lead time lengths are possible, we selected a weather-scale period that started 7 days before the event date (Webster et al., 2011). The seasonal scale-period started 6 months before the onset date and lasted until the start date of the weather-scale period so that the two periods do not overlap. The seasonal-scale period is split into 1, 3 and 6 month periods,

and the SPEIs with corresponding accumulation time periods were used.

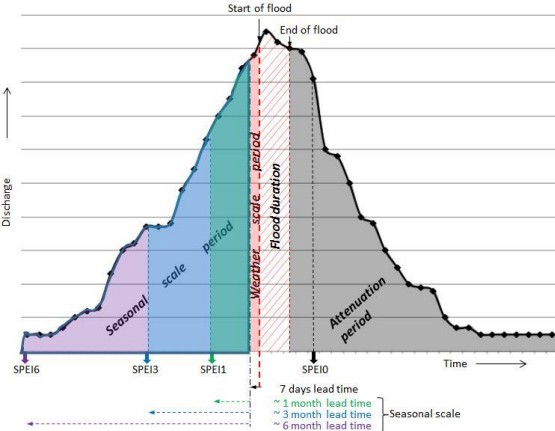

**Figure 2: Discharge during different time periods distinguished in data processing. The build-up period (until the start of the flood event) and its attenuation period were diagnosed from the increasing and decreasing sign of the**
**discharge, respectively. The start and the end date of the flood were derived from the NatCatSERVICE database (Munich Re, 2014). Weather scale period starts 7 days before the onset date of the event. The seasonal scale period starts 6 months before and continues until the start date of the flood event. The seasonal scale period is split in 1, 3 and 6 months periods.**

*Selecting reported floods*

The NatCatSERVICE data (Munich Re, 2014) includes two categories of inland flooding: a) riverine floods and b) flash floods. This study focused on riverine floods. Flash floods usually have a smaller extent, shorter build-up period and antecedent conditions play a less important role in their generation (Nied et al., 2014). We identified 501 damaging reported flood events in sub-Saharan Africa between 1980 and 2010. Some of the most catastrophic were the flood in Malawi in

March 1991, which resulted in 500 fatalities, the February 1999 and 2000 floods in Mozambique, which were responsible for the loss of 100 and 700 lives respectively, and the August 2006 flood in Ethiopia with 1000 reported fatalities.

Figure 3 shows the number of reported floods per year over the period 1980-2010 and the economic losses per year caused by these floods. There are several possible reasons for the upward trend, such as increased exposure due to population growth

and urbanization (Jongman et al., 2012) and underreporting of events in the earlier years due to limited penetration of communication technology (Kron et al., 2012).



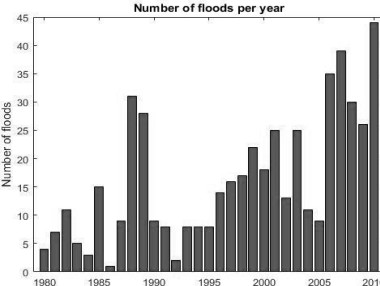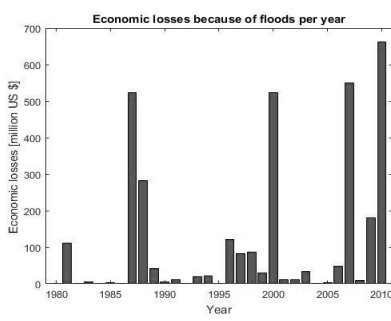

**Figure 3: Number of floods that are analysed in this study (left) and economic losses in million US $ per year caused by these floods (right) in sub-Saharan Africa between 1980 and 2010 (Munich Re, 2014).**
*7-day weather-scale lead precipitation*

Using the ERA Interim dataset, for each grid cell with a reported flood we calculated the accumulated 7-day running precipitation in the onset month of each flood for all 31 years of the dataset. This allowed us to obtain percentile categories per grid cell for this month: <33th percentile, 33rd to 66th percentile, >66th percentile. Subsequently, we divided the preceding accumulated 7-day precipitation of each flood event into the same categories. Due to uncertainty about the accuracy of the flood onset dates, we assumed that the real onset date could vary from the reported date up to three days before that date. For each of these three days we calculated the preceding 7-day precipitation and we retained the highest figure. Similarly, we calculated the mean monthly climatological precipitation values for each flooded grid cell over the 31 years of the dataset. Thus, for each of these selected cells, we sorted the months based on their climatological precipitation mean, discriminating between the six highest- and the six lowest-precipitation months.

*Seasonal-scale preceding SPEI values*

SPEI values for the months before a flood event are labelled SPEI1, SPEI3 and SPEI6, indicating lead time scales of 1, 3 and 6 months respectively (Figure 2). Although shorter-period SPEIs (e.g. SPEI1, 3) are part of the calculation of longer-period ones (e.g. SPEI6), their values can differ depending on the NP values of the other months. Hence, using SPEIs with different accumulation periods, we obtain updated seasonal-scale information related to floods. This is similar to the approach proposed by Goddard et al. (2014). Additionally, SPEI0, which refers to the end of the month that includes the flood's reported start date, was used as a reference to achieve a better understanding of SPEI changes over time. All month definitions were based on calendar days. Since floods may have occurred in different periods during a given calendar month, the actual length of the defined lead time varies by up to a month.

We first calculated the mean SPEI values of all reported flood events in the different time scales for an overview of the relation between the SPEI and the reported flood events. We then grouped the flood events geographically using spatial Food Production Units (FPUs). These are combined hydrological zones and economic regions as defined by Cai and Rosegrant, (2002) and updated by Kummu et al (2010). The FPUs are of comparable size and were chosen in lieu of countries, as they better depict river flood exposure (Jongman et al., 2012). In this way, we explored whether there are locations where the antecedent flood conditions played a more important role in flood generation than in others, exhibiting high SPEI values on the different time scales and identifying the geographical areas that demonstrated the wettest antecedent conditions on seasonal time scales. Finally, we identified and calculated the number of floods with exceptionally high wetness (SPEI > 1.5) over different seasonal time scales. Using a high threshold and identifying the number of no-flood cases that exceeded it during the 31 years of the dataset (false alarms) provides a better understanding of whether a high SPEI is applicable as a flood warning indicator.



*Combination of weather-scale precipitation and seasonal-scale antecedent conditions*

We divided the reported floods into groups according to the total 7-day lead precipitation, also identifying the extreme 7-day precipitation events. Hence, the groups created were: <33rd percentile, 33rd – 66th percentile, 66th – 99th percentile and >99th percentile. The mean SPEI value of the different time scales for each group was calculated. We thus evaluated the joint influence between seasonal-scale antecedent conditions and different magnitudes of weather-scale precipitation categories that led to a flood. Additionally, we identified the extreme 7-day precipitation events (above the 99th percentile) of the flooded grid cells during the onset month of each flood from 1980-2010 after which no-flood was reported and we compared their SPEIs with those of the weather-scale events of the same magnitude that led to flooding.

*Combination of seasonal-scale SPEIs with SPEI0 for floods and no-flood events*

In a final assessment, we combined seasonal-scale SPEIs (i.e. SPEI1, 3, and 6) with SPEI0. Taking into account all the flooded grid cells and the 31 years of the dataset, we compared the number of reported floods to the number of non-floods that exhibited similar SPEIs. Because the two periods are independent from each other, SPEI0 is not affected by the other SPEIs and, consequently, seasonal-scale periods could be combined with SPEI0 using the '&' condition (i.e. SPEI-6&0, SPEI-3&0, SPEI-1&0). The boundary values of the 'wet' categories were used as thresholds (i.e. 0, 0.5, 1, 1.5, 2). Comparing the number of floods to the number of non-floods for each threshold, we assess the likelihood that a high seasonal-scale SPEI combined with a high weather-scale SPEI0 has led to a flood event.

## 3 Results

### 3.1 Floods in sub-Saharan Africa

Figure 4a shows the spatial distribution of the 501 selected flood events over the period from 1980 to 2010. Floods were reported in most continental sub-Saharan countries. South Africa faced the highest number of reported flood events, followed by Kenya, Somalia, Mozambique and Ethiopia. Figure 4b shows the number of reported floods in sub-Saharan Africa per FPU. In southern Africa, a considerable number of floods were reported in the areas of the Limpopo and Zambezi river basins and along the coast of South Africa. Eastern Africa also experienced a significant number of flood events, mainly in the southern part of the Nile and near lakes Turkana and Victoria. Western Africa was divided into smaller FPUs than eastern Africa, and thus there were fewer floods reported in each FPU. Nevertheless, a concentration of floods was observed along the Volta, Niger and Senegal rivers. The pattern shows consistency with the floods reported by Dartmouth Flood Observatory (Global Archive of Large Flood Events, 2010, available at http://floodobservatory.colorado.edu/), which shows that most recent deadly floods happened in places where the population has increased more rapidly in recent years (Di Baldassarre et al., 2010).





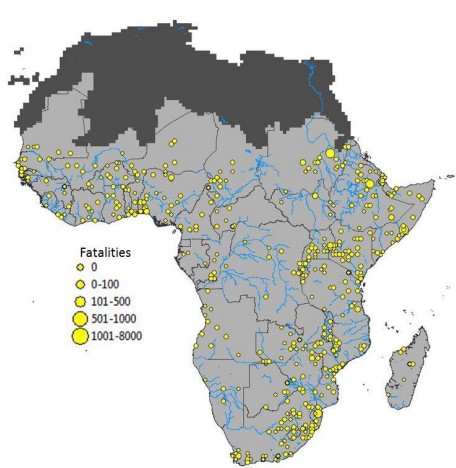

**Figure 4a Reported flood events (1980-2010) in sub-Saharan Africa grouped per country and major rivers. The size of the dots shows the fatalities that each flood caused (Munich Re, 2014).**

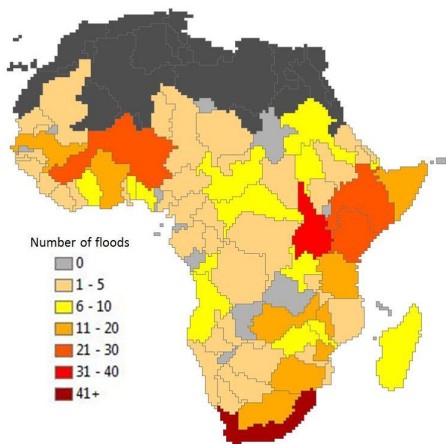

**Figure 4b Number of reported flood events per FPU (1980-2010) (Munich Re, 2014).**

### 3.2 Weather-scale results

10  Figure 5 (left) shows the reported floods categorized according to their accumulated 7-day lead precipitation. Since the local characteristics (location, timing, intensity) of each flood are different, the absolute amount of the 7-day precipitation can correspond to a different percentile category for each flood. For example, even though the total 7-day precipitation in a dry location is less than that in a wet location, the former will probably be positioned in a higher percentile category than the latter.

Weather-scale precipitation in the highest precipitation tercile ($> 66^{th}$) was associated with 72% of the floods, while for 21% of the reported floods, precipitation ranged between the $33^{rd}$ and $66^{th}$ tercile. The remaining percentage of floods was preceded by lead precipitation of the lowest tercile. This is in contrast to the NatCatSERVICE dataset (Munich Re, 2014), which nearly always indicates heavy or extreme rainfall just before the flood event as the main flood driving mechanism.



These differences could be explained, among other things, by the low quality of the reanalysis dataset in data-poor African countries, where observational data is rare. However, there may also be uncertainty in disaster databases regarding the exact location, the flood cause, and the accuracy of the onset date. Moreover, precipitation events can be very local and are often missed in the daily point-based data. Nevertheless, our results are similar to those of Douben (2006), who showed that the

frequency of heavy rain as a cause of flood damage in Africa is well above the global average (85%) for flood events during the years 1985 to 2003.

Figure 5 (right) shows in blue and red the floods that occurred during the six highest- and six lowest-precipitation months in each location, respectively. As expected, most floods happened during wet months. This also demonstrates a relationship

between flooding and rainfall. However, as the figure does not illustrate when exactly during the wet months the floods took place, we cannot conclude with certainty whether the antecedent conditions contributed to flood occurrence.

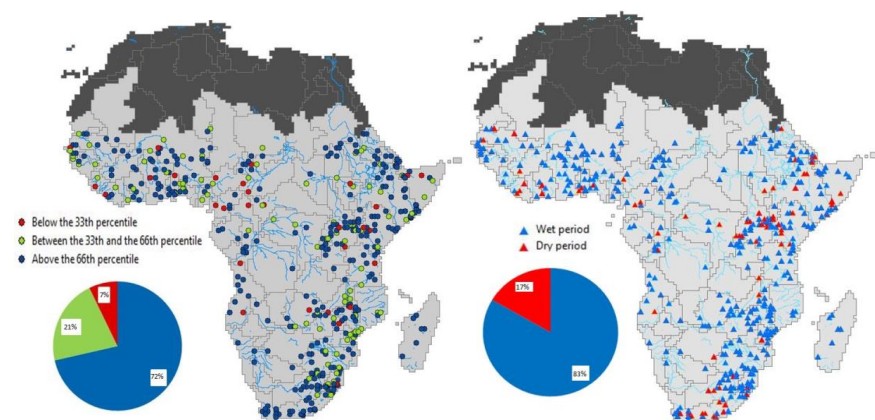

**Figure 5 Reported flood events on an FPU map of Africa. The different colors show the category (in percentiles) to which each flood belongs based on total 7-day antecedent precipitation (left) and during which period based on climatological mean precipitation each flood event took place (right).**

**3.3 Seasonal-scale results**

**3.3.1 SPEI and flood events**

Figure 6 shows the SPEI values of each flood event on different time scales (0, 1, 3, and 6 months prior to the recorded flood). SPEI0 captured quite well the wet conditions that developed during the month of the floods. More than one fourth of all reported floods (27%) coincided with an SPEI0 > 2, while 67% of all floods exhibited an SPEI > 1 (Figure 5a). Furthermore, 89% had an SPEI0 above the climatological mean, showing a high correlation between flooding and wetter-

than-average conditions.

A high percentage, 40%, of all floods was preceded by an SPEI1 in one of the three wettest categories (Figure 5b). At a 1-month time scale (SPEI1) the correlation with floods was clearly lower than that of SPEI0. However, a remarkable number of floods, 72%, had an SPEI1 above the climatological mean (above 0) and 57% above 0.5. SPEI3 and SPEI6 (Figure 5c and

5d) were characterized by a further reduction of the number of floods in the 'wet' categories. Although the percentage of floods with an SPEI > 0 remained the same, the number of floods with an SPEI3 > 0.5 was slightly above average (52%), while an increase in the number of floods in the drier categories was observed. In contrast to the previous SPEIs, an 'average wetness' category prevailed here (0 to 0.5), while in the previous ones the 'wetter' category was prevalent. SPEI6 was the




only time scale in which all wetness categories were part of the distribution pie. Excluding the extreme categories (both wet and dry), the rest were fairly well distributed.

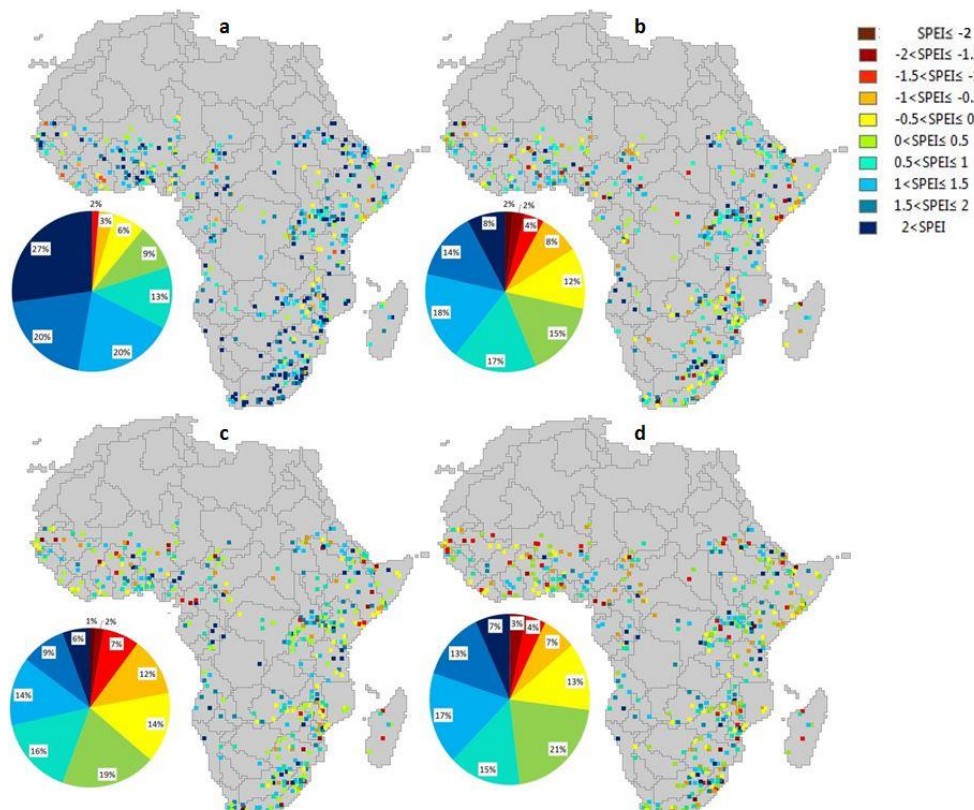

**Figure 6: Flood events per FPU for a) SPEI0, b) SPEI1, c) SPEI3, d) SPEI6. The different colors show the different SPEI categories as defined in Table 1. Dark blue is the wettest category and dark brown the driest.**

Figure 7 shows the mean SPEI of all reported floods. This value remains positive over the different time scales, indicating wetter than average conditions. Moreover, a smooth upward trend is observed as the flood event approaches, which becomes steep in the transition between mean SPEI1 (=0.6) and SPEI0 (=1.51), where the highest value is observed. The mean SPEI6 (0.26) is close to the average climatological conditions, while mean SPEI1 (=0.6) and SPEI3 (=0.54) exhibited comparable values and were positioned in the mild wet category (Table 1). The line shown is a smooth fit used for illustration purposes.

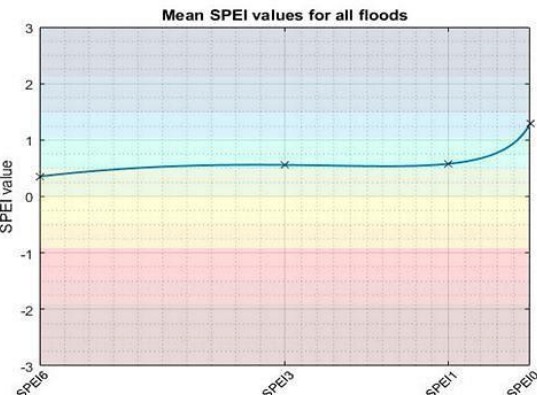

**Figure 7: Mean values of the considered SPEIs for all reported floods in sub-Saharan Africa.**

### 3.3.2 Classification in FPUs

The SPEIs of different time scales were also aggregated to FPU areas (Figure 8). Due to the small number of floods in many FPUs, only those with more than five reported floods are presented. Figure 8a shows clearly that in all cases the conditions during the flood's onset month were wetter than average. Southern Africa, which had the highest number of reported flood events, exhibited very high SPEI values (1.5 to 2). Some smaller FPUs in western Africa and the upper Nile area belonged to the same category. The FPU that exhibited the lowest values (0 to 0.5) was in the area of the Ivory Coast. Eastern Africa showed more homogeneous results since most of its FPUs exhibited SPEIs ranging from 1 to 1.5.

Some FPUs, such as those in the Horn of Africa, the Senegal region and Madagascar, exhibited consistently average wet or average dry conditions. The floods in these FPUs did not exhibit antecedent SPEI values that could be linked to flood signals. Instead, in these cases it seemed that flood events were correlated mainly with the short-term weather-scale phenomena during the flood's onset month. On the other hand, few FPUs had floods occurring with SPEI > 0.5 over all time scales. The FPU in Angola exhibited the highest SPEIs across the time scales. The FPU with the highest number of reported floods, at the coastline of South Africa, had an SPEI3 in a higher category than SPEI1, but its northern FPU (Limpopo area) showed the most consistent SPEI values over all seasonal time scales. In the Uganda area, where many floods were reported, we observed a gradual increase in the SPEI values, which may translate into an increasing flood probability. Although no strong consistency was observed on a wider geographical scale, the results showed that, up to 3 months before the flood, eastern Africa's FPUs exhibited higher SPEI values on a seasonal scale than those in western Africa. The small number of reported flood events per FPU, however, prevented statistical analyses that would show the areas where the local antecedent conditions generated the best flood signals.





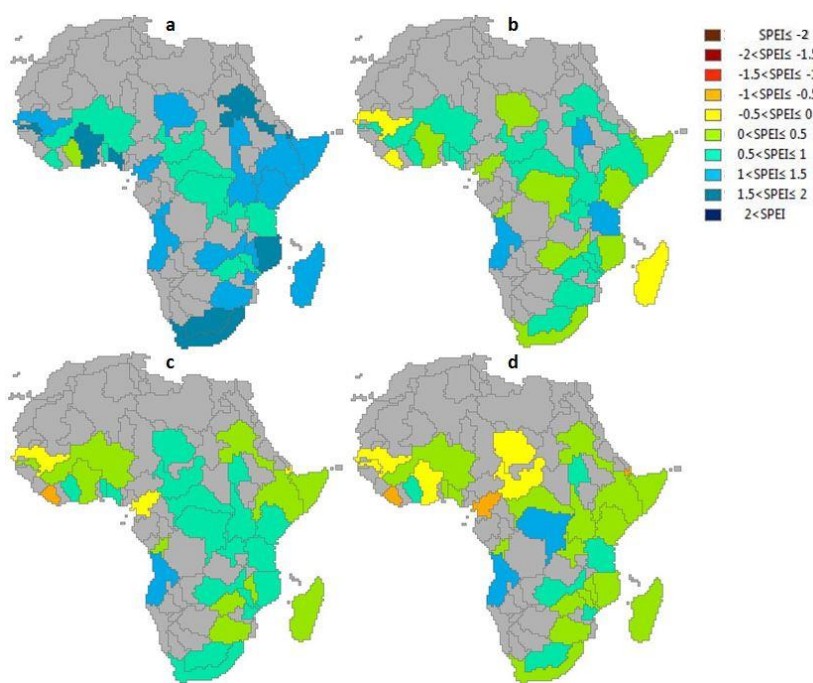

**Figure 8: Averaged SPEI values of flood events on an FPU level for the FPUs, where more than 5 floods are reported. a) SPEI0, b) SPEI1, c) SPEI3, d) SPEI6. The different colors show the different SPEI categories as defined in Table 1. Dark blue is the wettest category and dark brown the driest.**

**3.4 Combined weather-scale precipitation and SPEI results**

The mean SPEI values for the flood events grouped based on their 7-day lead precipitation percentile classes are shown in Figure 9 (see section 3.2). The extreme 7-day precipitation events (> 99[th] percentile) were also included, divided into those with a reported flood and those without.

Similarly to Figure 6, an upward trend is observed in the SPEI values of all categories as the disaster approaches. More specifically, it appeared that SPEI6 for all different flood categories was near the normal and mild wet categories, and SPEI3s and SPEI1s are all positioned in mild wet categories, except for the 33[rd] to 66[th] tercile, where there is a small decrease in SPEI1 compared to SPEI3. Finally, SPEI0 figures consistently reflected the differences in weather-scale precipitation, as they

followed the same order of the weather-scale precipitation categories (i.e. > 99[th] category had the highest SPEI0s and 0-33[rd] the lowest). However, the floods with extreme weather-scale precipitation (> 99[th] percentile) exhibited the lowest values in SPEI3. The way in which SPEI changes over time for the different precipitation categories gives indications that higher preceding wetness results in flooding at lower precipitation intensity values. Moreover, the positive SPEIs during all time scales indicate that the chance of having a flood during wetter than average climatological conditions increases the flood

likelihood. Nevertheless, the weather-scale processes during the onset month of each flood seemed to have contributed most to flood generation.

On the other hand, the seasonal-scale SPEI values of non-flood events with extreme weather-scale precipitation (>99[th] percentile) exhibited average to negative values, which shows that one of the possible reasons that no floods occurred could

be that dry antecedent conditions prevailed. Moreover, their mean SPEI0 is much lower than that of floods with the same

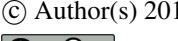


precipitation. This implies that the NP over the entire month was significantly higher in the flood cases than in the non-flood cases. Such non-flood cases during the 31 years are many and this justifies that their mean values are closer to the average conditions.

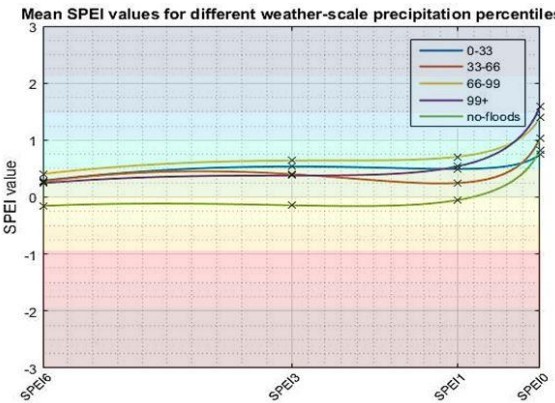

**Figure 9: Mean values of SPEIs for floods grouped based on their total 7-day precipitation percentiles ( 0-33rd, 33rd-66th, 66th-99th, ≥99th percentile) and no-flood cases with 7-day precipitation >99th.**

### 3.5 Seasonal SPEIs and SPEI0

Figure 10 shows the share of flood events compared to that of non-flood events for different combinations of seasonal-scale
SPEIs with SPEI0, using different SPEI threshold values. There is a steady increase in the number of flood events compared to the number of non-floods as the threshold increases. This shows that the higher the combined weather- and seasonal-scale SPEI values, the higher the flood likelihood. Particularly for the two highest thresholds, the percentage of flood events compared to non-floods is very high, indicating that when high SPEI is forecast or observed, this may be a reliable signal of flooding with a relatively small number of false alarms.

However, the heterogeneity of flood-producing mechanisms and the variety of the local boundary conditions (e.g. infrastructure of different safety levels) precludes setting a given SPEI value as a generic indicator of enhanced flood likelihood. A closer look at each flooded location is required. However, we may conclude that very high SPEI values (>1.5) in both weather- and seasonal-scale periods do lead to higher numbers of correct hits over false alarms, with a ratio close to 1.

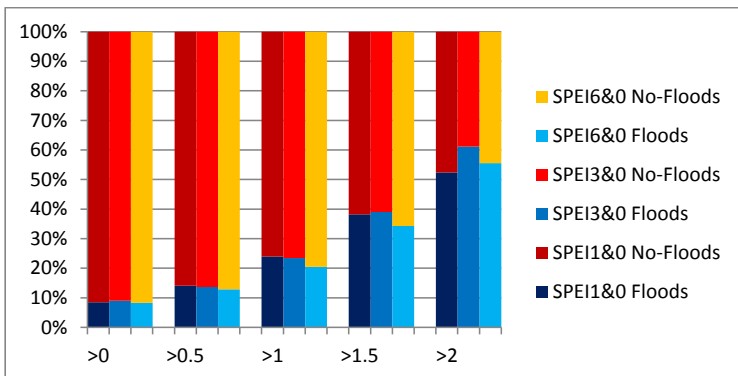

**Figure 10: Percentage of floods to no-floods under different thresholds and for different combinations of seasonal-scale SPEIs with SPEI0 .**



## 4 Discussion

*The role of short-term weather conditions*

The role of weather-scale meteorological conditions (particularly rainfall) in flood generation is generally accepted (Webster
et al., 2011; Jongman et al. 2014; Froidevaux et al., 2015). However, the length of the weather-scale period depends highly
on the local characteristics. For example, a 2-day precipitation sum is best correlated with flood frequency and magnitude in
the high ranges of the Swiss Alps, but longer-duration precipitation affects flood occurrence more in the in the western and
eastern Swiss Plateau (Froidevaux et al., 2015). Hence, we expect to be on the safe side by using a relatively long synoptic
time window (7 days) for large-scale floods. We included a range of three days prior to the flood onset date and performed
the weather-scale analysis over an area ($2.5° \times 2.5°$) around the reported flood event. The rationale behind this is to capture
the impact of the rainfall in neighbouring areas, including some upstream which, given the seven-day period, may have
contributed to the flood generation mechanisms. Nevertheless, we acknowledge that this is a simplified approach. Reality is
much more complicated, as the response of hydrological systems to precipitation varies considerably depending on time and
place (Eltahir and Yeh, 1999). Further studies should give this serious consideration, carrying out analyses on smaller spatial
scales and using hydrological models to estimate the travel and the concentration time of the upstream rainfall to each flood
location.

Despite the absence of high quality datasets in Africa (Rogers and Tsirkunov, 2013), the results showed a relationship
between the reported flood occurrence and lead precipitation (72% of all floods were preceded by precipitation positioned in
the highest tercile). This stresses the importance of weather scale forecasts in flood forecasting. However, an appreciable
number of floods were preceded by either average or low precipitation. This is in contrast with the qualitative additional
information of the disaster dataset, in which nearly all flood events were preceded by either high or extreme precipitation.
This difference can be attributed to either the limited quality of the reanalysis precipitation in the data-poor environment of
our study or to human error in the disaster dataset used. For instance, the characterization of a rainfall event as 'extreme' and
the reported flood onset date, especially in the 1980s, often seems to be arbitrary.

*Seasonal-scale factors*

Because research usually focuses on weather-scale events, the seasonal-scale antecedent conditions are often neglected
(Webster et al. 2011, Froidevaux et al. 2015). A range of different SPEI values reflected the antecedent conditions of the
reported floods on a seasonal scale. Although we acknowledge that each flood has unique characteristics, the mean SPEI of
all floods tended to increase over time while approaching the flood onset. At all seasonal time scales, the mean SPEI fell into
wet categories, showing that the wetness during the preceding months provides a flood signal. Moreover, the percentage of
floods that belong to the 'very wet' SPEI categories is higher than expected. As a standardized variable, the SPEI follows the
probability density function of the normal distribution. Hence, the probability of a location having a high SPEI, for instance
greater than 1.5, is 6.5%. Reported floods with such a high SPEI1 account for 22% of all floods. Most of these floods
exhibited either the highest or the second highest SPEI values during the 31 years of the dataset.

The classification of floods into areas of comparable size (i.e. FPUs) allowed for a comparison of the antecedent conditions
of different geographical areas. It should be noted, however, that most FPUs did not have a high number of flood events and
that the flood triggering patterns differ from place to place. In some cases, floods were reported without unusually wet
antecedent conditions. In other FPUs, (e.g. in the Uganda and Angola areas), the reported floods demonstrated high SPEIs
more consistently during the preceding months. The identification of areas with a higher chance of receiving flood signals is
an incentive to look at them in more detail.





*Events with 'strong' signals on a seasonal-scale*

Analysing flood events with very high SPEIs (>1.5) provides a better understanding of whether seasonal-scale warnings based on SPEI could be beneficial for flood identification. The fact that 14%, 19% and 21% of the total number of floods were linked to a very high SPEI (> 1.5), at 6-, 3- and 1-month periods before the reported onset month, respectively, gives an encouraging message that the forecasting uncertainty of some events on a seasonal scale could be decreased. The increasing percentage implies that floods become more likely at higher SPEI. In all three cases, approximately 60% high-SPEI events had either 0 or 1 false alarms, which means that the preceding conditions of these floods were the wettest over the 31 years.

The false alarm likelihood should be taken into account in the decision-making process for disaster preparedness activities if SPEI is used as a warning indicator. Quantifying the risk of false alarms in relation to the different SPEIs will be a subject for further research.

*Connection between seasonal- and weather-scale periods*

The connection of weather- with seasonal-scale conditions revealed that high wetness in the latter can increase the flood likelihood at lower weather-scale precipitation intensities. The mean seasonal-scale SPEI3 and SPEI6 values of the floods preceded by extreme lead precipitation (i.e. >99th percentile) are lower than those of flood events preceded by lower precipitation categories. Their mean SPEI0, however, has the highest value, indicating that these floods were mainly

generated by processes during the reported onset month. On the other hand, antecedent conditions become more important in the floods associated with precipitation events in the lower percentiles. For example, although the floods in the lowest weather-scale precipitation category (i.e. 0-33rd percentile) demonstrated the lowest SPEI0 values, they had higher seasonal-scale SPEIs than the extreme-precipitation floods. The category with the highest number of reported floods (66th-99th) also shows an upward trend of SPEIs, indicating that the increasing SPEI could be translated into higher flood likelihood.

However, although the floods with lead precipitation in the 33rd-66th percentile exhibited a relatively low SPEI0, their seasonal-scale SPEIs cannot explain their flood generation mechanism. Possible explanations are that these events were generated further upstream and are being routed to the downstream area or that they were induced by high-intensity rainfall at a low spatio-temporal resolution.

Our findings are in line with those of Berthet et al. (2009), who demonstrated that the variety in preceding moisture plays a major role in flood generation in France at similar levels of flood-triggering precipitation, and with Nied et al. (2014), who showed that a small amount of rainfall can result in flood generation when the soil is saturated. This conclusion is further supported by the fact that the vast majority of the reported floods (83%) were initiated during the wettest months in each location, where the hydro-meteorological phenomena before or during these months created flood-favourable conditions. The

combination of weather- and seasonal-scale condition is also supported by Pathiraja et al. (2012), who showed that there was an underestimation of the magnitude of flood flows in the Murray-Darling Basin in Australia when the joint influence of flood-producing rain events and antecedent wetness was not taken into consideration. Nevertheless, performing a more detailed analysis focusing on a (sub-)catchment area, including ground observations and the use of a hydrological model, could provide more detailed results.


Extreme weather-scale rainfall events are crucial in flood generation, especially in countries where flood protection safety levels are low. Nonetheless, they do not necessarily go hand-in-hand with floods. Although SPEI0 is higher when an extreme weather-scale precipitation event led to a reported flood than when it did not, the seasonal-scale SPEIs in these instances are very different. This could mean that a weather-scale rainfall forecast of an extreme event might not be highly relevant for



flood prediction when low preceding SPEI values are observed. Nevertheless, this is not the case for flash floods, which were not taken into account in this study.

Finally, combining the seasonal-scale SPEIs with SPEI0 indicated that a the percentage of floods over non-floods increases
with increasing SPEI thresholds, reaching a ratio of around 1 for SPEIs >2. Both floods and non-floods at this high SPEI value are apparently rare. Nevertheless, when such high SPEIs are correctly forecast, the likelihood of a false alarm is low. Nonetheless, we should acknowledge that each place where a flood was reported has different flood protections and thus different SPEIs may result in flooding. Finally, regarding the non-flood cases that exhibited very wet seasonal- and weather-scale conditions, it is not reasonable to declare with certainty that they were not floods since it there is a high probability that
they were simply not reported. As we have said, the false alarm likelihood must be taken into account in the decision-making process if the SPEI is to be used as a warning indicator.

*Reported floods*

Identifying the floods from the empirical disaster database (Munich Re, 2014) made it possible to take into account real flood events. The disaster datasets, however, are often susceptible to human errors and omissions (Jongman et al., 2015). For example, in the dataset used, there is an increasing trend in flood numbers over the years, which may be caused by an upward trend in reporting frequency rather than occurrence frequency. The Munich Re (2014) dataset, however, includes a pair of geographical coordinates of each flood event, which is lacking in other similar datasets (e.g. EM-DAT, DesInventar).

*Policy Relevance*

The approach fits well in the global policy on disaster management: the Sendai Framework of Disaster Risk Reduction (SFDRR) (UNISDR, 2015b). The Framework calls for enhanced efforts to reduce risk from natural hazards (including
floods), such as protection, financial risk transfer and early warning systems (Mysiak et al., 2016).. Seasonal forecasting systems, are promising measures that complement existing warning systems, and support post disaster risk reduction strategies such as relief operations. For this, the SPEI-based approach of using seasonal information to prepare for flood events could be further developed and eventually used to support disaster preparedness activities in the regions at risk. For example, a Forecast-based Financing (FbF) approach is currently being developed by the Climate Centre of the Red
Cross/Red Crescent (Coughlan De Perez et al., 2015). It aims to disburse humanitarian funding based on forecast information, with the ultimate goal of making disaster risk management as effective as possible. It involves disaster preparedness measures and humanitarian actions that can be implemented in the time between the warning forecast and the disaster. Another recently developed method that fits well with our seasonal-scale approach is the 'Ready-Set-Go' concept (Goddard et al., 2014). In this approach, each phase of disaster preparedness is activated when the output of different forecast
types (e.g. seasonal, weather), exceeds a certain threshold. In this case, such a threshold could be based on SPEI values as presented in this paper.

## 5 Conclusions

The framework developed in this paper explores the influence of antecedent conditions on reported flood events in sub-
Saharan Africa for the period 1980-2010. Our analysis included 501 large-scale reported floods based on the Munich Re disaster database (Munich Re, 2014). In contrast to the majority of previous studies, we have clearly distinguished the antecedent conditions between weather and seasonal scales based on their reported onset date: (a) the weather-scale conditions encompass 0-7 days prior to each flood event and are captured by the 7-day accumulated precipitation, (b) the





seasonal-scale conditions are reflected in values of the Standardized Precipitation Evapotranspiration Index (SPEI) 1, 3 and 6 months before each flood.

The results indicate that the vast majority of floods (72%) received 7 days lead precipitation that is positioned in the highest
tercile (>66[th]), relative to the local conditions of the flood's onset month. Furthermore, the mean SPEI0 (1.51) of all floods is higher than the mean seasonal-scale SPEIs (0.6, 0.54, 0.26 for SPEI1, SPEI3 and SPEI6 respectively). These outcomes demonstrate the catalytic role of hydro-meteorological phenomena in flood generation during the days close to the flood's onset, emphasizing the importance of weather forecasts in flood forecasting.

At the seasonal scale, the SPEI values of each flood 1, 3 and 6 months before the month of the flood's onset were calculated. The mean SPEI value of all floods is positive over all time scales, indicating wetter-than-average conditions before the events, and an upward trend is observed as the flood approaches. Nevertheless, there is a considerable number of floods that did not exhibit positive SPEIs on a seasonal scale. Grouping the flood events according to Food Production Units (FPUs), we observed that flood events in some areas showed higher SPEI values on a seasonal-scale than others. This shows that the
antecedent conditions in some areas play a more important role in flood generation than in others. Nonetheless, due to the small number of floods in each FPU area, no statistically robust conclusions could be drawn. Regarding the cases with very high seasonal-scale conditions, a considerable number of floods exhibited severely wet conditions (i.e. SPEI > 1.5), most of which (around 60%) had the highest values during the 31 years of the dataset. Nevertheless, a detailed hydrological analysis for each individual flood location would help determine accurate thresholds if SPEI is used as a flood warning indicator. We
thus suggest that future research focus on such small spatial scales in order to estimate the local flood risk.

The combined analysis of weather- and seasonal-scale flood antecedent conditions reveals that their joint influence affects flood generation. The flood events preceded by extreme weather-scale precipitation, in several cases, exhibited lower SPEI values (i.e. drier conditions) than the floods preceded by less intensive precipitation. Moreover, false alarms decrease in
proportion to increasing SPEI thresholds of reported floods. Translating the results into practice, we conclude that the magnitude of seasonal-scale wetness could be a useful input to the weather-scale flood forecasts based on which disaster actions are to be taken. This information could be related to ongoing work at the IFRC or other relief organizations to better prepare relief actions. Another promising approach, which could use SPEI threshold values as an input is the 'Ready-Set-Go' concept, where each disaster preparedness and relief phase is activated when the output of different forecast types (e.g.
seasonal or weather) exceeds a pre-set threshold.

### Acknowledgments

We thank Munich Re for providing reported flood data from the NatCatSERVICE database. The project was funded by *NWO VICI Grant nr.* 016.140.067, and NWO grant nr. 869-15 001.

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
