# Peer review of "The influence of antecedent conditions on flood risk in sub-Saharan Africa"

_Natural Hazards and Earth System Sciences, 2017_

## Referee Comment (RC1) · Anonymous Referee #1 · 24 Feb 2017

The authors investigate the influence of high-intensity rain events and antecedent moisture conditions on flood probability in a large target area, including almost the entire African continent. Based on a data set of reported floods (provided by Munich RE), the short-term (event-precipitation) and long-term conditions (SPEI) before each event are systematically compared. The results indicate, that most of the reported floods are related to high precipitation events during the last seven days. Further the authors argue, that rather moist conditions on seasonal scale lead to enhanced flood risk, most likely due to filled up storage systems. While the research target is timely and the manuscript is well structured and easy to follow, I have some serious concerns about the statistical methods and the interpretation of the result. Particularly the conclusions remain very

vague! Thus I recommend to extend the statistical analysis and to better test, whether the conclusions are robust and really supported by the underlying data sets.

In the following I will summarize my major concerns. Since I expect the text to change significantly, I will not go into detail at this stage of the review process.

1) Introduction and data sets: The presentation of previous research in the field of flood forecasting in Africa is very short. No information is given on the timing of floods in different subbasins of this vast target area and on the general climatic conditions. This information would be highly valuable in order to interpret (and scrutinize) the results of the statistical analysis. (E.g. It could be interesting to identify some differences between Eastern and Western Africa, which seem to behave differently in terms of the SPEI-flood relationship.) Likewise the introduction of the data sets is insufficient. I would expect a detailed description of the advantages and drawbacks of the data – especially the daily precipitation values on a coarse grid are extremely uncertain, since they do not cover local scale convective events, which frequently trigger high-intensity rain. The Munich Re data are introduced in two sentences only – again I think it would make sense to better discuss its origin and shortcomings!

2) Statistical Methods: The majority of the methods used is purely descriptive and does not allow to draw verified conclusions. One example is Fig. 9., which shows the mean seasonal SPEI values before reported floods. The authors argue, that floods, which are not preceded by high-intensity rains (0-33%-interval) have larger SPEI-values during antecedent seasons. However, all of the lines are very close to each other. A test for statistical significance (t-test or similar) would be necessary to support this statement. Further a presentation with boxplots would be more suitable, since it does not only show the mean, but also the range (and overlap) of the different classes.

3) Dependency of SPEI and 7-day precipitation An increased SPEI value before flood events might have different reasons. One would be the limited capacity of the storage. A second one could be the persistence of the climate (if it has been moist for the last

couple of months, a high-intensity rain event might be more probable). That would explain, why the 66-99% interval in Fig. 9 shows the highest SPEI for all seasons. In order to draw robust conclusions, it would be necessary to disange those processes. In Fig. 10, the frequency of flood under different SPEI combinations is shown. The analysis of further combinations (e.g. SPEI0 → normal and SPEI3 → moist) could support the conclusions of the manuscript. Again, a test of statistical significance is highly recommended.

Would it further be possible to show a point-cloud of seasonal SPEI against 7-day-precipitaion for all flood events? A clear negative relationship (higher SPEI values lead to flooding although 7-day precipitation is not extreme), would also support the conclusion.

4) The authors highlight, that a forecast based on the findings is possible and that uncertainties could be reduced. I have the feeling that this is very optimistic. Would It be possible to establish a very simple tool for each of the FPU-units (e.g. based on a SPEI threshold value) and quantify the probability of hits and false alarms?

5) Conclusions and discussion: The conclusions and the discussion section include many statements, which are not proven by the data or by literature (E.g. second paragraph, page 15, but also others). I recommend to carefully check, and to focus on findings which are really supported by the data.

Minor remarks: 1) Section 2.1 and 2.2 could have more meaningful subtitles. 2) p.5l.5: The weather scale and SPEI periods do not overlap and the SPEI period lasts until the date of the weather scale period. This is not possible, since the SPEI is defined on a monthly time scale? Does the SPEI period end with the month before the flood event? 3) Fig. 2: I am confused about the hydrograph. Is this a schematic figure or is it somehow based on discharge time series? If I understand the figure correctly, discharge already increases seasons in advance. Usually the start of a flood event is defined as the first significant increase of discharge (which would be 5 months in

advance in Fig. 2). 4) p6.l14: Floods are grouped into wet and dry seasons? How exactly is this relevant for the statistical analysis?

---

## Referee Comment (RC2) · Anonymous Referee #2 · 26 Apr 2017

Dear authors and editors,

I evaluated this paper exploring the use of SPEI and 7-days antecedent precipitation as indicators of damage triggering floods in the sub-Saharan Africa.

If I put the glasses and look at the manuscript in the viewpoint of an NGO looking for an assessment about this topic, then I would be rather satisfied with this report.

As contribution for the scientific community this manuscript: - lacks of rigorous description of the data sources and their limitations; - makes in my opinion wrong use of the term "lead time" in many sections; - has a rather small sample; - do not looks at missed events; - presents a very simplistic descriptive statistical evaluation; - poorly

acknowledges recent effort in seasonal forecasting (e.g. http://www.hydrol-earth-syst-sci.net/special_issue824.html).

Concerning the missed events, have you tried to obtain information about events not-reported in the Münich-RE report, but being taxed as potential flooding in FPU with less than 5 events?

I am generally very positive with respect to pragmatic approaches like this, but here I have the feeling that here more efforts are needed in order to better support the statements concerning the potential of this method as a early indicator of floods.

Please consider also the comments in the PDF

Best regards

Please also note the supplement to this comment:
http://www.nat-hazards-earth-syst-sci-discuss.net/nhess-2017-58/nhess-2017-58-RC2-supplement.pdf

---

## Author Comment (AC1) · 7 Jun 2017

GENERAL COMMENT: The authors investigate the influence of high-intensity rain events and antecedent moisture conditions on flood probability in a large target area, including almost the entire African continent. Based on a data set of reported floods (provided by Munich RE), the short-term (event-precipitation) and long-term conditions (SPEI) before each event are systematically compared. The results indicate, that most of the reported floods are related to high precipitation events during the last seven days. Further the authors argue, that rather moist conditions on seasonal scale lead to enhanced flood risk, most likely due to filled up storage systems. While the research

target is timely and the manuscript is well structured and easy to follow, I have some serious concerns about the statistical methods and the interpretation of the result. Particularly the conclusions remain very vague! Thus I recommend to extend the statistical analysis and to better test, whether the conclusions are robust and really supported by the underlying data sets. In the following I will summarize my major concerns. Since I expect the text to change significantly, I will not go into detail at this stage of the review process.

RESPONSE: We thank the reviewer for his/her comments, and we are pleased that he/she finds the topic timely and the paper easy to follow. Based on the comments and suggestions, we suggest a thorough revision of the original manuscript. The revised version of the paper includes an extended statistical analysis to support our conclusions. At the end of this response, we present the additional new figures proposed to be included in the revised manuscript and we look forward to other comments. Below, we address the comments point-by-point. Please note that the figures are in a pdf file in the supplement.

COMMENT 1: Introduction and data sets: The presentation of previous research in the field of flood forecasting in Africa is very short. No information is given on the timing of floods in different sub-basins of this vast target area and on the general climatic conditions. This information would be highly valuable in order to interpret (and scrutinize) the results of the statistical analysis. (E.g. It could be interesting to identify some differences between Eastern and Western Africa, which seem to behave differently in terms of the SPEI-flood relationship.) Likewise the introduction of the data sets is insufficient. I would expect a detailed description of the advantages and drawbacks of the data – especially the daily precipitation values on a coarse grid are extremely uncertain, since they do not cover local scale convective events, which frequently trigger high-intensity rain. The Munich Re data are introduced in two sentences only – again I think it would make sense to better discuss its origin and shortcomings!

RESPONSE 1: We would like to thank Reviewer#1 for his/her comments and recommendations. In the revised version, we include the Köppen climatological map (Level 1) to give more information regarding the general climatic conditions of each flooded location (Figure 1). Moreover, we compare the 3 main climatological areas (see Figure 2) in terms of their SPEIs. Although some differences in terms of SPEI-flood relationship are observed, these are not statistically significant.

Furthermore, we do agree that the datasets lack a rigorous description. Therefore, in the revised manuscript we discuss:

a) The uncertainty in disaster datasets, and the reasons for the discrepancies between them (e.g. different entry criteria, time period covered) b) An extended description of the sources of NatCatSERVICE database. c) The possible explanations of the upward trend in reported floods over time. d) The difference between the hydrological definition of a flood compared to the definition that it is used by flood loss reporting, and discuss how this difference can be associated with missed events. e) The uncertainty of hydrological variables in the reanalysis dataset due to the lack of ground-based precipitation records in our area of interest. f) The sensitivity of the reanalysis product to the resolution choice. g) The uncertainty in hydrological variables in tropical regions and in southern Africa h) The large uncertainty in of the daily precipitation reanalysis dataset in capturing local-scale high intensity precipitation events.

COMMENT 2: Statistical Methods: The majority of the methods used is purely descriptive and does not allow to draw verified conclusions. One example is Fig. 9., which shows the mean seasonal SPEI values before reported floods. The authors argue, that floods, which are not preceded by high-intensity rains (0-33%-interval) have larger SPEI-values during antecedent seasons. However, all of the lines are very close to each other. A test for statistical significance (t-test or similar) would be necessary to support this statement. Further a presentation with boxplots would be more suitable, since it does not only show the mean, but also the range (and overlap) of the different classes.

RESPONSE 2: We agree with Reviewer #1 that the methods used in the original manuscript are rather descriptive. To accommodate the comments of Reviewer #2, we performed in preparation of the revised manuscript a statistical comparison between Flood and No-Flood events, presented by means of boxplots (see Figures 3, 4). These figures show the SPEI values of all flood and no-flood events on different time scales (0, 1, 3 and 6 months prior to the recorded flood). For each 'flooded cell' the no-flood cases that are taken into account refer to the particular flood onset month of the no-flood years. Due to the very high number of no-flood events, the median value is close to 0 at all time-scales. The SPEI0-SPEI6 values of floods are significantly higher, which is underpinned by the results of a z-test (p=0,05). More specifically, the median value of SPEI0, exhibits a value close to 1. This indicates that, as expected, the wetness in the end of these months was high, demonstrating that SPEI0 could be used as a flood monitoring tool. On the (overlapping) seasonal time scales we see a positive relationship between reported floods and SPEI, which reduces while moving from SPEI1 to SPEI6 (0.5 to 0.1).

Regarding the different weather-scale intervals (i.e. 7-day precipitation terciles): their median SPEI values of the different 7-day precipitation terciles exhibit any statistical significance differences and therefore in the original manuscript, we used descriptive results. In the revised version, we have removed this Figure (Figure 8 in the original manuscript (SPEI per precipitation class). Instead, we compare the maximum 7-day precipitation during no-flood events in the 31-year record with the 7-day precipitation that preceded the flood events. For each flood, we standardized the 31 values (1 for the Flood (F) and the 30 for No Floods (NF), with a mean of 0 and standard deviation of 1 (see Figure 5). This figure presents in boxplots the standardized 7-day precipitation (PRE7) of Flood (F) and No-Floods (NF) events. The results of the z-test showed that preceding PRE7 of floods did not exhibit any significant difference with that of no-floods (p> 0,1). This shows that although PRE7 that preceded the flood is high, it does not fully justify the flood generation: There probably were also similar magnitude events, in the same locations and during the same months that floods were reported, that did not

lead to a (reported) flood.

Being aware of the dataset limitations (e.g. incapacity of reanalysis datasets to capture convective rainfall events, likely inaccurate onset date, etc.), the message that we want to convey is that since we observe a relation between seasonal SPEI and flooding and that relation does not need a relation via weather-scale precipitation, implies that there probably is another factor that affects flooding on a seasonal scale prior to flood generation. This factor is assumed to be the soil saturation due to limited water storage capacity. These factors have been addressed in the revised version of the paper.

COMMENT 3: Dependency of SPEI and 7-day precipitation. An increased SPEI value before flood events might have different reasons. One would be the limited capacity of the storage. A second one could be the persistence of the climate. That would explain, why the 66-99% interval in Fig. 9 shows the highest SPEI for all seasons. In order to draw robust conclusions, it would be necessary to disengage those processes. In Fig. 10, the frequency of flood under different SPEI combinations is shown. The analysis of further combinations (e.g. SPEI0 - normal and SPEI3 - moist) could support the conclusions of the manuscript. Again, a test of statistical significance is highly recommended. Would it further be possible to show a point-cloud of seasonal SPEI against 7-dayprecipitaion for all flood events? A clear negative relationship (higher SPEI values lead to flooding although 7-day precipitation is not extreme), would also support the conclusion.

RESPONSE 3: As also mentioned in the response of Comment 2, we observed small but non-significant differences of the SPEI values between the precipitation classes. Therefore, we have removed this section from the analysis. In order to evaluate the persistence of the climate, we use the Pearson coefficient (R) between SPEI0 and PRE7 with seasonal SPEIs (Table 1). The results show that there is no significant correlation, which implies that there is no apparent climate persistence. This lack of correlation implies that the positive seasonal SPEIs (as shown in Figures 3, 4), did not affect the conditions during the flood onset month, but they played a role in flood

generation.

Moreover, in the revised version, we compare the frequency of floods and no-floods under different SPEI combinations. Since our sample size is not large, the combination of different categories of SPEI (e.g. SPEI0-normal & SPEI3-high) gives us a small number of events for each one of them. Therefore, we have used exceedance thresholds (e.g. SPEI0>0 & SPEI3>2). Based on the frequency of flood and no-flood events, we quantify the elevated probability of having a flood event compared to a no-flood under each combination (Figure 7). Using higher thresholds, the seasonal SPEIs behave differently. However, only few floods exceeded these thresholds (i.e. less than 10) and consequently, we can't draw any firm conclusions. For lower thresholds, we observe significantly elevated probabilities of having a flood. For example, when SPEI0>0 and SPEI1>1, it is 4 times more likely to have a flood event than can be expected assuming a random flood generation process.

COMMENT 4: The authors highlight, that a forecast based on the findings is possible and that uncertainties could be reduced. I have the feeling that this is very optimistic. Would It be possible to establish a very simple tool for each of the FPU-units (e.g. based on a SPEI threshold value) and quantify the probability of hits and false alarms?

RESPONSE 4: We agree that establishing a forecast-based financing system based on the finding of this research is quite a challenge. We have formulated our recommendations more cautiously. Our conclusions now include a message that forecast based financing could be based on some results of our research, and that there are seasonal flood signals, which might support more effective forecast-based risk mitigation solutions.

Regarding the FPU-units, we have decided to leave them out of this paper, as their sample size is small. Instead, we have identified statistically significant differences in larger areas, with different climatological characteristics . Regarding the very simple tool that Reviewer #1 mentions, we include a graph that shows the elevated probability, which is calculated by the frequency of floods and no-floods that exceed several thresholds (see R.3 and Figure 6, 7).

COMMENT 5) Conclusions and discussion: The conclusions and the discussion section include many statements, which are not proven by the data or by literature (E.g. second paragraph, page 15, but also others). I recommend to carefully check, and to focus on findings which are really supported by the data.

RESPONSE 5: In the revised manuscript, we now make statements that they are based on statistical analysis and have supported our conclusions with extended literature research.

Minor remarks

COMMENT 6: Section 2.1 and 2.2 could have more meaningful subtitles.

RESPONSE.6: We agree and have changed these titles.

COMMENT 7: p.5l.5: "The weather scale and SPEI periods do not overlap and the SPEI period lasts until the date of the weather scale period." This is not possible, since the SPEI is defined on a monthly time scale? Does the SPEI period end with the month before the flood event?

RESPONSE 7: We understand that in this part clarification is needed. SPEI is indeed defined on a monthly time-scale, but the seasonal SPEI period (SPEI1 , SPEI3, SPEI6) ends on the month before the month that the 7-day period is calculated. For example, if a flood is reported on January 1st, the 7-day period ends on December 23rd and the SPEI periods ends in November. In the revised text we have explained this more clearly.

COMMENT 8: Fig. 2: I am confused about the hydrograph. Is this a schematic figure or is it somehow based on discharge time series? If I understand the figure correctly, discharge already increases seasons in advance. Usually the start of a flood event is defined as the first significant increase of discharge (which would be 5 months in

advance in Fig. 2).

RESPONSE 8: Indeed, this graph might be confusing. Its is mainly to provide the reader a better understanding of the different time periods. The continuously increasing high discharge during the antecedent months does not represent reality. We have replaced it with a new one, where the discharge has a significant increase close to flood onset date.

COMMENT 9: p6.l14: Floods are grouped into wet and dry seasons? How exactly is this relevant for the statistical analysis?

RESPONSE 9: We agree with Reviewer #1 that the grouping the floods in wet and dry seasons does not provide any extra data for the statistical analysis. So, it is removed from the revised manuscript.

Additional References

In the revised document, we aim to include the following references;

Dutra, E., Di Giuseppe, F., Wetterhall, F., and Pappenberger, F.: Seasonal forecasts of droughts in African basins using the Standardized Precipitation Index, Hydrol. Earth Syst. Sci., 17, 2359-2373, doi:10.5194/hess-17-2359-2013, 2013 Stephens, E., J. J. Day, F. Pappenberger, and H. Cloke (2015),Precipitation and floodiness, Geophys. Res. Lett., 42, 10,316–10,323, doi:10.1002/ 2015GL066779 Coughlan de Perez, E., Stephens, E., Bischiniotis, K., van Aalst, M., van den Hurk, B., Mason, S., Nissan, H., and Pappenberger, F.: Should seasonal rainfall forecasts be used for flood preparedness?, Hydrol. Earth Syst. Sci. Discuss., doi:10.5194/hess-2017-40, in review, 2017 Seibert, M., Merz, B., and Apel, H.: Seasonal forecasting of hydrological drought in the Limpopo Basin: a comparison of statistical methods, Hydrol. Earth Syst. Sci., 21, 1611-1629, doi:10.5194/hess-21-1611-2017, 2017 Zhang, Y., Moges, S., and Block, P.: Does objective cluster analysis serve as a useful precursor to seasonal precipitation prediction at local scale? Application to western Ethiopia, Hydrol. Earth

[Figure]

Syst. Sci. Discuss., doi:10.5194/hess-2017-70, in review, 2017 Tschoegl L., Below R. and Guha‐Sapir D. (2006). An Analytical review of selected data sets on natural disasters and impacts. Paper prepared for the UNDP/CRED Workshop on Improving Compilation of Reliable Data on Disaster Occurrence and Impact. Bangkok, 2‐4 April 2008. Below R, Wirtz, A. and Guha-Sapir D. (2009), Disaster category classification and peril terminology for operational purposes. Paper prepared fror CRED and Munich Re, October 2009. Lorenz, C., and H. Kunstmann (2012), The Hydrological cycle in three state-of-the-art reanalyses: Intercomparison and performance analysis,J. Hydrometeorol.,13(5), 1397–1420, doi:10.1175/JHM-D-11-088.1 Herold, N., A. Behrangi, and L. V. Alexander (2017), Large uncertainties in observed daily precipitation extremes over land, J. Geophys. Res. Atmos., 122, 668–681, doi:10.1002/2016JD025842. Zhan, W., K. Guan, J. Sheffield, and E. F.Wood (2016), Depiction of drought oversub-Saharan Africa using reanalyses precipitation data sets,J. Geophys. Res.Atmos.,121, 10,555–10,574, doi:10.1002/2016JD02485). 8 Zhang, Q., H. Körnich, and K. Holmgren (2013), How well do reanalyses represent the southern African precipitation?,Clim. Dyn.,40(3–4),951–962, doi:10.1007/s00382-012-1423-z

Please also note the supplement to this comment:
http://www.nat-hazards-earth-syst-sci-discuss.net/nhess-2017-58/nhess-2017-58-AC1-supplement.pdf

**Supplement:**

**New Figures and tables**

[Figure]

Figure 1 Floods locations in sub-Saharan Africa from 1980 to 2010.
Background color shading refers to Köppen climate classes
(Green: Tropical climate; Brown: Oceanic Climate; Yellow: Dry Climate)

[Figure]

Figure 2 SPEI values in different sub-Saharan climatic zones, identified in Figure 1 (T-Tropical, D-Dry climate, O-Oceanic climate). On each box, the red line is the median, the edges of the box are the 25th and 75th percentiles, the whiskers extend to the most extreme and the outliers are plot individually. If the notches in the box plots do not overlap, you can conclude, with 95% confidence, that the true medians do differ.

[Figure]

SPEI values for Flood (F) and No-Flood (NF) events

[Figure]

Figure 3 Histogram and fitted distribution of flood events (red line)
compared to the fitted distribution of the no-flood events (magenta line).
The blue bars show the frequency of the flood events in each SPEI class

[Figure]

Figure 4: 7-day precipitation (PRE7) that preceded the flood events (F) and maximum 7-day precipitation for the No-Flood events (NF)

[Figure]

Figure 5 Times more likely to flood when SPEIs exceed a threshold (x axis)

[Figure]

Figure 6 Times more likely to flood when SPEIs exceed a threshold (x axis)

[Figure]

Figure 7 SPEIs per precipitation percentile

---

## Author Comment (AC2) · 7 Jun 2017

Reviewer #2: Approach of potential interest but currently lacking significant statistical evidence

GENERAL COMMENT: Dear authors and editors, I evaluated this paper exploring the use of SPEI and 7-days antecedent precipitation as indicators of damage triggering floods in the sub-Saharan Africa. If I put the glasses and look at the manuscript in the viewpoint of an NGO looking for an assessment about this topic, then I would be rather satisfied with this report. As contribution for the scientific community this manuscript: - lacks of rigorous description of the data sources and their limitations;

[Figure]

- makes in my opinion wrong use of the term "lead time" in many sections; - has a rather small sample; - do not looks at missed events; - presents a very simplistic descriptive statistical evaluation; - poorly acknowledges recent effort in seasonal forecasting (e.g. http://www.hydrol-earth-systsci.net/special_issue824.html). Concerning the missed events, have you tried to obtain information about events not reported in the Münich-RE report, but being taxed as potential flooding in FPU with less than 5 events? I am generally very positive with respect to pragmatic approaches like this, but here I have the feeling that here more efforts are needed in order to better support the statements concerning the potential of this method as a early indicator of floods. Please consider also the comments in the PDF.

RESPONSE: We thank the reviewer for his/her comments, and we are pleased that he/she is very positive about pragmatic approaches like this. Upon his/her comments, we have thoroughly revised the paper. The revised manuscript in preparation includes an extended statistical analysis to support our conclusions. We have also addressed the limitations of our study that have been mentioned by the reviewer, and have removed some of our results, which could not be supported by statistical analysis. At the end of this response, we present the additional new figures proposed to be included in the revised manuscript and we look forward to other comments. Below, we address the comments point-by-point. Please note that the new figures are in pdf file in the supplement.

COMMENT 1: The manuscript lacks of rigorous description of the data sources and their limitations.

RESPONSE 1: We thank Reviewer #2 for his/her comment and after re-reading the manuscript, we agree that the strengths and limitations of the data sources were not presented thoroughly. In the revised version we have addressed the following descriptions;

a) The uncertainty in disaster datasets, and the reasons for the discrepancies between

them (e.g. different entry criteria, time period covered) b) An extended description of the sources of NatCatSERVICE database. c) The possible explanations of the upward trend in reported floods over time. d) The difference between a hydrological flood event and a reported flood event as listed in the Munich RE database, and how this is associated with missed events. e) The uncertainty of hydrological variables in the reanalysis dataset due to the lack of ground-based precipitation records, especially in developing countries. f) The sensitivity of the reanalysis product in the resolution choice. g) The uncertainty in hydrological variables in tropical regions and in southern Africa h) The large uncertainty of daily precipitation reanalysis due to the incapacity of capturing local-scale high intensity precipitation events.

COMMENT 2: It makes in my opinion wrong use of the term "lead time" in many sections.

RESPONSE 2: Yes, that is correct. The term 'lead time' is associated with forecasts, while this paper examines the conditions prior to the flood events. We have replaced 'lead time' with 'antecedent time', throughout the revised paper.

COMMENT 3: It has a rather small sample. COMMENT 4: It does not look at missed events. COMMENT 5: Concerning the missed events, have you tried to obtain information about events not reported in the Münich-RE report, but being taxed as potential flooding in FPU with less than 5 events?

RESPONSE 3, 4, 5: We agree that the sample used in our study is rather small. To increase the sample size, we also included flood events reported in the earlier years of the dataset (i.e. from the 1980s onwards). Regarding the missed events, we would like to emphasize that NatCatSERVICE database does not include a flood based on the hydrological definition (i.e. high water levels, peak discharges). Instead, an event enters the dataset when there is property damage and/or when there are people affected. Hence, the paper focuses only on these damaging events, which are usually the ones that humanitarian organizations are interested in. However, by examining only the grid

points where floods were reported and not all the grid points of sub-Saharan Africa, we have decreased the number of missed events. In the revised version, we explicitly mention these assumptions. Finally, when revising our paper, we have omitted analyses based on regional FPUs, since we feel that our dataset is too limited to identify enough data points per FPU.

COMMENT 6: It presents a very simplistic descriptive statistical evaluation.

RESPONSE 6: We agree that the original version of the paper presented a descriptive statistical rather than inferential statistical evaluation. In the revised version, we now perform statistical significance tests, which are carried out for;

a) The SPEIs of flood and no-flood events (figures 1, 2) b) The 7-day precipitation of flood and no-flood events (Figure 3) c) Both 7-day precipitation and SPEIs across different climatological areas (Figure 4) d) The SPEIs of different 7-day precipitation percentiles (Figure 5)

Figures 1 and 2 show the SPEI values of all flood and no-flood events on different time scales (0, 1, 3 and 6 months prior to the recorded flood). The no-flood cases considered refer, for each 'flooded cell', to the particular flood onset month of the no-flood years. Due to the very high number of no-flood events, the median value at all time scales is close to 0. The SPEI0-SPEI6 median values of floods are significantly higher, which is underpinned by the results of the z-test ($p=0.05$). More specifically, the median value of SPEI0, exhibits a value close to 1. This indicates that, as expected, the wetness in the end of these months was high, demonstrating that SPEI0 could be used as a flood monitoring tool. On the (overlapping) seasonal time scales we see a positive relationship between reported floods and SPEI, which reduces while moving from SPEI1 to SPEI6 (0.5 to 0.1).

In addition, we compare the maximum 7-day precipitation of each location during the no-flood years to the 7-day precipitation that preceded the flood events. For each flood, we standardized the 31 values over our 31 year of data (1 for the Flood (F) and the 30

for No Floods (NF), with a mean of 0 and standard deviation of 1 (see Figure 3). This figure presents in boxplots the standardized 7-day precipitation (PRE7) of Flood (F) and No-Floods (NF) events. The results of the z-test showed that preceding PRE7 of floods did not exhibit any significant difference with that of no-floods ($p= 0.1$). This shows that although PRE7 that preceded the flood is high, it does not fully explains the flood generation. There were probably also similar magnitude events, in the same locations and during the same months that floods were reported, that did not lead to a (reported) flood.

Being aware of the dataset limitations (e.g. incapacity of reanalysis datasets to capture convective rainfall events, likely inaccurate onset date, etc.), the message that we want to convey is that since we observe a relation between seasonal SPEI and flooding and that this relation does not need a relation via weather-scale precipitation, implies that there probably is another factor that affects flooding on a seasonal scale prior to flood generation. One factor might be the soil saturation due to limited water storage capacity. This has been discussed in the revised version of the paper.

Finally, we have omitted the original Figure 8 (SPEI per precipitation class) in the revised version, since conclusions based on this figure could not be supported with statistical significant differences.

COMMENT 7: It poorly acknowledges recent effort in seasonal forecasting (e.g. http://www.hydrol-earth-systsci.net/special_issue824.html).

RESPONSE 7: We thank Reviewer #2 for the link. In the revised version, we include some of these references (see 'Additional References').

Comments in the PDF

COMMENT 8: Consequently, they are more reliant on post-disaster response and preparedness activities, often assisted by international donors and humanitarian organizations. Consider here to have a look at Stephens et al. "Floodiness" approach.

Stephens, E., J. J. Day, F. Pappenberger, and H. Cloke (2015), Precipitation and flood-iness, Geophys. Res. Lett., 42, 10,316–10,323, doi:10.1002/2015GL066779.

RESPONSE 8: In the revised version, we have included this reference.

COMMENT 9: Here you might have a look at a recent HEPEX-HESS special issue on sub-seasonal to seasonal forecasting http://www.hydrol-earth-syst-sci.net/special_issue824.html RESPONSE 9: We have read some interesting and relevant papers of the HEPEX-HESS special issue and we have included some of them in the revised version (see 'Additional References').

COMMENT 10: The long-term ('seasonal-scale') wetness reflected in the SPEI for the preceding 1, 3 and 6 months, and (b) the short-term ('weather-scale') cumulative rainfall over the 7 days preceding the event. You could call this also long-memory and short memory disposition. RESPONSE 10: Thank you for your remark. The text of the revised version has been changed significantly and we have taken this remark into consideration.

COMMENT 11: Figure 1: Schematic overview of the approach followed in this study. %False alarms seems a quite trivial metric for a sound assessment. RESPONSE 11: We agree.

COMMENT 12: Munich Re NatCatSERVICE disaster database. Is there any cross-validation of the accuracy/completeness of this data source? RESPONSE 12: Unfortunately, we have not conducted any cross-validation of NatCatSERVICE database. The reason is that it is the only dataset at our disposal that provides details of reported flood events, such as coordinates, onset and end dates for the entire sub-Saharan Africa since 1980. We have also looked at other disaster datasets such as EM-DAT and DesInventar, but a systematic cross-validation was not possible as they did not have detailed geographical descriptions, and have only a limited number of reported floods. In the revised version, we recommend to conduct a cross-validation in future research when new reporting data will emerge.
COMMENT 13: How you deal with the mismatch between the 0.5° and the 2.5° resolution? Wouldn't TRMM be an option to evaluate recent years? RESPONSE 13: We think that the datasets should be consistent in their spatial scale and therefore in the revised version, we have upscaled the 0.5° to 2.5° resolution, in order to take into account a larger flood affected area. TRMM provides observations only since 2000 and therefore we'd rather not use it for the sake of consistency throughout the paper.

COMMENT 14: A statistical procedure was applied to fit the accumulated records to a log-logistic distribution with a mean of 0 and a standard deviation of 1. This should be shown or referenced RESPONSE 14: We agree. In the methods sections of the revised manuscript we include the relevant references.

COMMENT 15: Above you speak about 7 days preceding the event, and here you speak about 7 days "lead time". In my understanding lead time is associated to forecasts RESPONSE 15: That is correct. We agree that the 7 days preceding the event would rather be called "antecedent time" and not "lead time". We now only refer to 'lead time' when we talk about forecasts.

COMMENT 16: Figure 2: In this sketch I have the impression that you are dealing with a rather trivial problem, since 7 days prior to the flood peak you have already about 50% of the flood volume and 90% of the peak level. Please discuss. Minor: Add also SPEI0, Minor: Think about the word "lead time" RESPONSE 16: We thank reviewer #2 for this remark. We agree that the graph is confusing. The purpose of this graph is mainly to provide the reader a better understanding of the time points of SPEI and 7-day precipitation. The figure doesn't show any real flood event. We agree that usually a flood is defined with a significant discharge increase close the flood onset. In the revised manuscript, we replaced the graph with a more realistic one. Moreover, SPEI0 is shown on the map. Finally, we have substituted 'lead time' with 'antecedent time'.

COMMENT 17: How efficient is the flood reporting for each country? RESPONSE 17: Unfortunately, to our knowledge, there is not any research that analyzes the efficiency

of the flood reporting in each African country.

COMMENT 18: High correlation between flooding and wetter-than-average conditions. (trivial) RESPONSE 18 Indeed, we agree that this sentence does not give any important information to the results and we have removed it.

COMMENT 19: In how many cases was SPEI1> 1 but no flood was recorded? RESPONSE 19: Taking into account all the 'flooded cells' and the months that in each one the flood was generated, we get 1731 cases with SPEI1>1 that no flood was recorded. This number accounts for 11.5% of all no-flood cases. In the revised version, we compare the percentages of flood and no-flood events that exceeded different thresholds and we quantify the elevated probability found for flood events (see Figure 6).

COMMENT 20: SPEI3 > 0.5 was slightly above average (52%), Is my understanding correct: If SPEI3 is > 0.5 then in about 50% of the cases you might expect a flood. Is this not very close to throwing a coin? RESPONSE 20: Our dataset consists of 501 floods over 31 years. Every flood is placed on a grid cell and therefore for each grid cell there is 1 flood and 30 no floods in the record. Therefore, there are 15030 cases of no-floods. SPEI0>1 for 1666 no-flood cases (11%), SPEI1>1 for 1731 (11.5%) no-flood cases (11.5%), SPEI3>1 for 1571 no-flood cases (10.5%) and SPEI6>1 for 1454 no-flood cases (9.5%), while the corresponding percentages for floods is 41% (SPEI0), 27% (SPEI1), 21.5% (SPEI3), 16% (SPEI6). In the revised version, we present these increased probabilities (see Figure 6). Moreover, in the revised manuscript, we also combine the seasonal SPEIs with SPEI0 and 7-day precipitation. Following the same way of thinking, comparing the percentage of floods and no-floods that exhibited an SPEI3>0.5, we are arguing that it is twice more likely to have a flood in the location, where a flood was reported, when SPEI3> 0.5.

COMMENT 21: Fig.7 Why not showing S here? RESPONSE 21: We have used boxplots in the revised version of our paper (e.g. see Figures 1, 3, 4 and 5).

COMMENT 22: Limpopo basin - Check the references Seibert, M., Merz, B., and Apel,

H.: Seasonal forecasting of hydrological drought in the Limpopo Basin: a comparison of statistical methods, Hydrol. Earth Syst. Sci., 21, 1611-1629, doi:10.5194/hess-21-1611-2017, 2017. Trambauer, P., Werner, M., Winsemius, H. C., Maskey, S., Dutra, E., and Uhlenbrook, S.: Hydrological drought forecasting and skill assessment for the Limpopo River basin, southern Africa, Hydrol. Earth Syst. Sci., 19, 1695-1711, doi:10.5194/hess-19-1695-2015, 2015. Dutra, E., Di Giuseppe, F., Wetterhall, F., and Pappenberger, F.: Seasonal forecasts of droughts in African basins using the Standardized Precipitation Index, Hydrol. Earth Syst. Sci., 17, 2359-2373, doi:10.5194/hess-17-2359-2013, 2013. RESPONSE 22: We thank Reviewer #2 for these relevant and interesting articles. In the revised version, we have included include some of them (see 'Additional References').

COMMENT 23: (> 99th percentile) How many samples are in each 7-day precipitation category for the reported flood events? RESPONSE 23: The samples in each 7-day category are: 0-33 percentile: 53 cases, 33-66 percentile: 119 cases, 66-100 percentile: 329 cases.

COMMENT 24: Fig9. 5-colored boxplots welcome RRESPONSE 24: We have included boxplots in our figures.

COMMENT 25: Fig10 I try to understand, If SPEI1&0 are above 2, then in about 50% of the cases a flood occurred. Correct? RESPONSE 25: Yes, that is correct. However, as explained in R.12, the no-flood cases are way more than the flood cases. In the revised version, we include Figure 7, in which we show the elevated probability of having a flood when SPEI0 & SPEI1>0.

Additional References

In the revised document, we aim to include the following references;

Dutra, E., Di Giuseppe, F., Wetterhall, F., and Pappenberger, F.: Seasonal forecasts of droughts in African basins using the Standardized Precipitation Index, Hydrol.

[Figure]

Earth Syst. Sci., 17, 2359-2373, doi:10.5194/hess-17-2359-2013, 2013 Stephens, E., J. J. Day, F. Pappenberger, and H. Cloke (2015),Precipitation and floodiness, Geophys. Res. Lett., 42, 10,316–10,323, doi:10.1002/ 2015GL066779 Coughlan de Perez, E., Stephens, E., Bischiniotis, K., van Aalst, M., van den Hurk, B., Mason, S., Nissan, H., and Pappenberger, F.: Should seasonal rainfall forecasts be used for flood preparedness?, Hydrol. Earth Syst. Sci. Discuss., doi:10.5194/hess-2017-40, in review, 2017 Seibert, M., Merz, B., and Apel, H.: Seasonal forecasting of hydrological drought in the Limpopo Basin: a comparison of statistical methods, Hydrol. Earth Syst. Sci., 21, 1611-1629, doi:10.5194/hess-21-1611-2017, 2017 Zhang, Y., Moges, S., and Block, P.: Does objective cluster analysis serve as a useful precursor to seasonal precipitation prediction at local scale? Application to western Ethiopia, Hydrol. Earth Syst. Sci. Discuss., doi:10.5194/hess-2017-70, in review, 2017 Tschoegl L., Below R. and Guha‐Sapir D. (2006). An Analytical review of selected data sets on natural disasters and impacts. Paper prepared for the UNDP/CRED Workshop on Improving Compilation of Reliable Data on Disaster Occurrence and Impact. Bangkok, 2‐4 April 2008. Below R, Wirtz, A. and Guha-Sapir D. (2009), Disaster category classification and peril terminology for operational purposes. Paper prepared fror CRED and Munich Re, October 2009. Lorenz, C., and H. Kunstmann (2012), The Hydrological cycle in three state-of-the-art reanalyses: Intercomparison and performance analysis,J. Hydrometeorol.,13(5), 1397–1420, doi:10.1175/JHM-D-11-088.1 Herold, N., A. Behrangi, and L. V. Alexander (2017), Large uncertainties in observed daily precipitation extremes over land, J. Geophys. Res. Atmos., 122, 668–681, doi:10.1002/2016JD025842. Zhan, W., K. Guan, J. Sheffield, and E. F.Wood (2016), Depiction of drought oversub-Saharan Africa using reanalyses precipitation data sets,J. Geophys. Res.Atmos.,121, 10,555–10,574, doi:10.1002/2016JD02485). 8 Zhang, Q., H. Körnich, and K. Holmgren (2013), How well do reanalyses represent the southern African precipitation?,Clim. Dyn.,40(3–4),951–962, doi:10.1007/s00382-012-1423-z
Please also note the supplement to this comment:
http://www.nat-hazards-earth-syst-sci-discuss.net/nhess-2017-58/nhess-2017-58-AC2-supplement.pdf
* * *
[Figure]

**Supplement:**

**New Figures**

[Figure]

Figure 1 SPEI values for Flood (F) and No-Flood (NF) events

[Figure]

Figure 2 Histogram and fitted distribution of flood events (red line) compared to the fitted distribution of the no-flood events (magenta line). The blue bars show the frequency of the flood events in each SPEI class

[Figure]

Figure 3: 7-day precipitation (PRE7) that preceded the flood events (F)
and maximum 7-day precipitation for the No-Flood events (NF)

[Figure]

Figure 4 SPEI values in different sub-Saharan climatic zones, identified in Figure 1 (T-Tropical, D-Dry climate, O-Oceanic climate). On each box, the red line is the median, the edges of the box are the 25th and 75th percentiles, the whiskers extend to the most extreme and the outliers are plot individually. If the notches in the box plots do not overlap, you can conclude, with 95% confidence, that the true medians do differ.

[Figure]

Figure 5 SPEIs per precipitation percentile

[Figure]

Figure 6 Times more likely to flood when SPEIs exceed a threshold (x axis)

[Figure]

Figure 7 Times more likely to flood when SPEIs exceed a threshold (x axis)

[Figure]

Figure 8 Floods locations in sub-Saharan Africa from 1980 to 2010.
Background color shading refers to Köppen climate classes
(Green: Tropical climate; Brown: Oceanic Climate; Yellow: Dry Climate)

---

## Author Response (AR1)

**List of changes compared to original manuscript**

Big changes in the text (abstract introduction methodology discussion conclusions)

|    | big changes in the text (abstract, introduction, includelogy, inscussion, conclusions)                                                                                                                                                                          |
|----|-----------------------------------------------------------------------------------------------------------------------------------------------------------------------------------------------------------------------------------------------------------------|
| 5  |  <li>Extra references about;</li> <li>a) Data limitations</li> <li>b) Advances in flood forecasting Africa</li> <li>c) Advances on seasonal forecasting</li>                                                                                           |
| 10 |  <li>Changes in the methodology</li> <li>a) Changes in weather-scale precipitation section</li>                                                                                                                                                        |
|    |  <li>Removal of the precipitation terches section, since no significant differences were found</li> <li>Standardization and comparison of the maximum 7-day precipitation of No-Floods to the preceding
7-day precipitation of floods (PRE7)</li>  |
| 15 | • Standardization and comparison of the max 7-day precipitation (MAX7) of the flood onset months to the PRE7 and to the maximum 7-day precipitation of No-Floods.                                                                                               |
|    | b) Changes in seasonal-scale section                                                                                                                                                                                                                            |
| 20 | Removal of the FPU section                                                                                                                                                                                                                                      |
| 20 | Comparison of flood to no-flood SPEIs     Palative odd calculation for SPEI thrashold exceedance                                                                                                                                                                |
|    | · Kelative oud calculation for 51 Ef threshold exceedance                                                                                                                                                                                                       |
|    | c) Changes in combined weather- and seasonal-scale section                                                                                                                                                                                                      |
| 25 | Removal of False Alarms graph                                                                                                                                                                                                                                   |
| 25 | Relative odds when combining seasonal SPEIs and SPEI0                                                                                                                                                                                                           |
|    | Keiative odds when combining seasonal SPEIS and PKE/                                                                                                                                                                                                            |
|    |                                                                                                                                                                                                                                                                 |

**Reviewer 1**

30

35

GENERAL COMMENT: The authors investigate the influence of high-intensity rain events and antecedent moisture conditions on flood probability in a large target area, including almost the entire African continent. Based on a data set of reported floods (provided by Munich RE), the short-term (event-precipitation) and long-term conditions (SPEI) before each event are systematically compared. The results indicate, that most of the reported floods are related to high precipitation events during the last seven days. Further the authors argue, that

- rather moist conditions on seasonal scale lead to enhanced flood risk, most likely due to filled up storage systems. While the research target is timely and the manuscript is well structured and easy to follow, I have some serious concerns about the statistical methods and the interpretation of the result. Particularly the conclusions
  remain very vague! Thus I recommend to extend the statistical analysis and to better test, whether the conclusions are robust and really supported by the underlying data sets. In the following I will summarize my major concerns. Since I expect the text to change significantly, I will not go into detail at this stage of the review
- major concerns. Since I expect the text to change significantly, I will not go into detail at this stage of the review process.
   RESPONSE: We thank the reviewer for his/her comments, and we are pleased that he/she finds the topic timely

45 RESPONSE: We thank the reviewer for his/her comments, and we are pleased that he/she finds the topic timely and the paper easy to follow. Based on the comments and suggestions, we suggest a thorough revision of the original manuscript. Below, we address the comments point-by-point.

- COMMENT 1: Introduction and data sets: The presentation of previous research in the field of flood forecasting in Africa is very short. No information is given on the timing of floods in different sub-basins of this vast target area and on the general climatic conditions. This information would be highly valuable in order to interpret (and scrutinize) the results of the statistical analysis. (E.g. It could be interesting to identify some differences between Eastern and Western Africa, which seem to behave differently in terms of the SPEI-flood relationship.) Likewise the introduction of the data sets is insufficient. I would expect a detailed description of the advantages and
- 55 drawbacks of the data especially the daily precipitation values on a coarse grid are extremely uncertain, since they do not cover local scale convective events, which frequently trigger high-intensity rain. The Munich Re data are introduced in two sentences only – again I think it would make sense to better discuss its origin and shortcomings!
- 60 RESPONSE 1: We would like to thank Reviewer#1 for his/her comments and recommendations. In the revised version, we include the Köppen climatological map (Level 1) to give more information regarding the general

climatic conditions of each flooded location. Moreover, we have compared different climatological and geographical areas in terms of their SPEIs. Although some differences in terms of SPEI-flood relationship are observed, these are not statistically significant. Therefore, we have not drawn any conclusions out of that. Furthermore, we do agree that the datasets lack a rigorous description. Therefore, in the revised manuscript we

- 5 discuss: a) The uncertainty in disaster datasets, and the reasons for the discrepancies between them (e.g. different entry criteria, time period covered) b) An extended description of the sources of NatCatSERVICE database. c) The possible explanations of the upward trend in reported floods over time. d) The difference between the hydrological definition of a flood compared to the definition that it is used by flood loss reporting, and discuss how this difference can be associated with missed events. e) The uncertainty of hydrological variables in the
- 10 reanalysis dataset due to the lack of ground-based precipitation records in our area of interest. f) The sensitivity of the reanalysis product to the resolution choice. g) The uncertainty in hydrological variables in tropical regions and in southern Africa h) The large uncertainty in of the daily precipitation reanalysis dataset in capturing local-scale high intensity precipitation events.
- 15 COMMENT 2: Statistical Methods: The majority of the methods used is purely descriptive and does not allow to draw verified conclusions. One example is Fig. 9., which shows the mean seasonal SPEI values before reported floods. The authors argue, that floods, which are not preceded by high-intensity rains (0-33%-interval) have larger SPEI-values during antecedent seasons. However, all of the lines are very close to each other. A test for statistical significance (t-test or similar) would be necessary to support this statement. Further a presentation with boxplots would be more suitable, since it does not only show the mean, but also the range (and overlap) of the
- different classes.

RESPONSE 2: We agree with Reviewer #1 that the methods used in the original manuscript are rather descriptive. To accommodate the comments of Reviewer #1, we performed a statistical comparison between

- 25 Flood and No-Flood events, presented by means of boxplots. For each 'flooded cell' the no-flood cases that are taken into account refer to the particular flood onset month of the no-flood years. Due to the very high number of no-flood events, the median value is close to 0 at all time-scales. The SPEIO-SPEI6 values of floods are significantly higher, which is underpinned by the results of a z-test (p=0,05). More specifically, the median value of SPEI0, exhibits a value close to 1. This indicates that, as expected, the wetness in the end of these months was high demonstrating that SPEI0 could be used as a flood monitoring tool. On the (overlapping) seasonal time.
- 30 high, demonstrating that SPEI0 could be used as a flood monitoring tool. On the (overlapping) seasonal time scales we see a positive relationship between reported floods and SPEI, which reduces while moving from SPEI1 to SPEI6 (0.5 to 0.1). Regarding the different weather-scale intervals (i.e. 7-day precipitation terciles): their median SPEI values of the different 7-day precipitation terciles do not exhibit any statistical significance differences and therefore in the original manuscript, we used descriptive results. In the revised version, we have
- 35 removed this part. Instead, we compare the maximum 7-day precipitation during no-flood events in the 31-year record with the 7-day precipitation that preceded the flood events. For each flood, we standardized the 31 values (1 for the Flood (F) and the 30 for No Floods (NF), with a mean of 0 and standard deviation of 1. The results of the z-test showed that preceding PRE7 of floods did not exhibit any significant difference with that of no-floods (p> 0,1). This shows that although PRE7 that preceded the flood is high, it does not fully justify the flood
- generation: There probably were also similar magnitude events, in the same locations and during the same months that floods were reported, that did not lead to a (reported) flood. Being aware of the dataset limitations (e.g. incapacity of reanalysis datasets to capture convective rainfall events, likely inaccurate onset date, etc.), the message that we want to convey is that since we observe a relation between seasonal SPEI and flooding and that relation does not always need a relation via weather-scale precipitation, implies that there probably is another
   factor that affects flooding on a seasonal scale prior to flood generation. This factor could be the soil saturation
- due to limited water storage capacity. These factors have been addressed in the revised version of the paper.

COMMENT 3: Dependency of SPEI and 7-day precipitation. An increased SPEI value before flood events might have different reasons. One would be the limited capacity of the storage. A second one could be the persistence
 of the climate. That would explain, why the 66-99% interval in Fig. 9 shows the highest SPEI for all seasons. In order to draw robust conclusions, it would be necessary to disengage those processes. In Fig. 10, the frequency of flood under different SPEI combinations is shown. The analysis of further combinations (e.g. SPEI0 - normal and SPEI3 - moist) could support the conclusions of the manuscript. Again, a test of statistical significance is highly recommended. Would it further be possible to show a point-cloud of seasonal SPEI against 7-

55 dayprecipitation for all flood events? A clear negative relationship (higher SPEI values lead to flooding although 7-day precipitation is not extreme), would also support the conclusion.

RESPONSE 3: As also mentioned in the response of Comment 2, we observed small but non-significant differences of the SPEI values between the precipitation classes. Therefore, we have removed this section from the analysis. In the revised version, we compare the frequency of floods and no-floods under different SPEI

combinations. Since our sample size is not large, the combination of different categories of SPEI (e.g. SPEI0normal & SPEI3-high) gives us a small number of events for each one of them. Therefore, we have used exceedance thresholds (e.g. SPEI0>0 & SPEI3>2). Based on the frequency of flood and no-flood events, we quantify the elevated probability of having a flood event compared to a no-flood under each combination. For

example, when SPEID>0 and SPEI1>1, it is 4 times more likely to have a flood event than can be expected 5 assuming a random flood generation process.

COMMENT 4: The authors highlight, that a forecast based on the findings is possible and that uncertainties could be reduced. I have the feeling that this is very optimistic. Would It be possible to establish a very simple tool for each of the FPU-units (e.g. based on a SPEI threshold value) and quantify the probability of hits and false alarms?

RESPONSE 4: We agree that establishing a forecast-based financing system based on the finding of this research is quite a challenge. We have formulated our recommendations more cautiously. Our conclusions now include a message that forecast based financing could be based on some results of our research, and that there are seasonal flood signals, which might support more effective forecast-based risk mitigation solutions. Regarding the FPU-units, we have decided to leave them out of this paper, as their sample size is small, not allowing us to draw robust conclusions. Regarding the very simple tool that Reviewer #1 mentions, we include graphs that show the elevated probability, which is calculated by the frequency of floods and no-floods that exceed several

20 thresholds (see R.3).

10

15

25

COMMENT 5) Conclusions and discussion: The conclusions and the discussion section include many statements, which are not proven by the data or by literature (E.g. second paragraph, page 15, but also others). I recommend to carefully check, and to focus on findings, which are really supported by the data.

RESPONSE 5: In the revised manuscript, we now make statements that they are based on statistical analysis and have supported our conclusions with extended literature research.

**Minor remarks 30**

COMMENT 6: Section 2.1 and 2.2 could have more meaningful subtitles.

RESPONSE.6: We agree and have changed these titles.

- COMMENT 7: p.51.5: "The weather scale and SPEI periods do not overlap and the SPEI period lasts until the 35 date of the weather scale period." This is not possible, since the SPEI is defined on a monthly time scale? Does the SPEI period end with the month before the flood event?
- RESPONSE 7: We understand that in this part clarification is needed. SPEI is indeed defined on a monthly time-40 scale, but the seasonal SPEI period (SPEI1, SPEI3, SPEI6) ends on the month before the month that the 7-day period is calculated. For example, if a flood is reported on January 1st, the 7-day period ends on December 23rd and the SPEI periods ends in November. In the revised text we have explained this more clearly.
- COMMENT 8: Fig. 2: I am confused about the hydrograph. Is this a schematic figure or is it somehow based on 45 discharge time series? If I understand the figure correctly, discharge already increases seasons in advance. Usually the start of a flood event is defined as the first significant increase of discharge (which would be 5 months in advance in Fig. 2).
- RESPONSE 8: Indeed, this graph might be confusing. Its is mainly to provide the reader a better understanding 50 of the different time periods. The continuously increasing high discharge during the antecedent months does not represent reality. We have replaced it with a new one, where the discharge has a significant increase close to flood onset date.

COMMENT 9: p6.114: Floods are grouped into wet and dry seasons? How exactly is this relevant for the statistical analysis?

RESPONSE 9: We agree with Reviewer #1 that the grouping the floods in wet and dry seasons does not provide any extra data for the statistical analysis. So, it is removed from the revised manuscript.

**Reviewer #2**

"Approach of potential interest but currently lacking significant statistical evidence"

- 5 GENERAL COMMENT: Dear authors and editors, I evaluated this paper exploring the use of SPEI and 7-days antecedent precipitation as indicators of damage triggering floods in the sub-Saharan Africa. If I put the glasses and look at the manuscript in the viewpoint of an NGO looking for an assessment about this topic, then I would be rather satisfied with this report. As contribution for the scientific community this manuscript: - lacks of rigorous description of the data sources and their limitations; - makes in my opinion wrong use of the term "lead
- 10 time" in many sections; has a rather small sample; do not looks at missed events; presents a very simplistic descriptive statistical evaluation; poorly acknowledges recent effort in seasonal forecasting (e.g. http://www.hydrol-earth-systsci.net/special\_issue824.html). Concerning the missed events, have you tried to obtain information about events not reported in the Münich-RE report, but being taxed as potential flooding in FPU with less than 5 events? I am generally very positive with respect to pragmatic approaches like this, but
- 15 here I have the feeling that here more efforts are needed in order to better support the statements concerning the potential of this method as a early indicator of floods. Please consider also the comments in the PDF.

RESPONSE: We thank the reviewer for his/her comments, and we are pleased that he/she is very positive about pragmatic approaches like this. Upon his/her comments, we have thoroughly revised the paper. The revised manuscript includes an extended statistical analysis to support our conclusions. We have also addressed the like the paper with the there is the there is the paper.

limitations of our study that have been mentioned by the reviewer, and have removed some of our results, which could not be supported by statistical analysis. Below, we address the comments point-by-point.

COMMENT 1: The manuscript lacks of rigorous description of the data sources and their limitations.

- RESPONSE 1: We thank Reviewer #2 for his/her comment and after re-reading the manuscript, we agree that the strengths and limitations of the data sources were not presented thoroughly. In the revised version we have addressed the following descriptions; a) The uncertainty in disaster datasets, and the reasons for the discrepancies between them (e.g. different entry criteria, time period covered) b) An extended description of the
- 30 sources of NatCatSERVICE database. c) The possible explanations of the upward trend in reported floods over time. d) The difference between a hydrological flood event and a reported flood event as listed in the Munich RE database, and how this is associated with missed events. e) The uncertainty of hydrological variables in the reanalysis dataset due to the lack of ground-based precipitation records, especially in developing countries. f) The sensitivity of the reanalysis product in the resolution choice. g) The uncertainty in hydrological variables in
- 35 tropical regions and in southern Africa h) The large uncertainty of daily precipitation reanalysis due to the incapacity of capturing local-scale high intensity precipitation events.

COMMENT 2: It makes in my opinion wrong use of the term "lead time" in many sections. RESPONSE 2: Yes, that is correct. The term 'lead time' is associated with forecasts, while this paper examines the conditions prior to the flood events. We have replaced 'lead time' with 'antecedent time', throughout the revised paper.

COMMENT 3: It has a rather small sample.

20

25

40

COMMENT 4: It does not look at missed events.

COMMENT 5: Concerning the missed events, have you tried to obtain information about events not reported in the Münich-RE report, but being taxed as potential flooding in FPU with less than 5 events?

RESPONSE 3, 4, 5: We agree that the sample used in our study is rather small to produce statistically robust results. However, to our knowledge, this study is unique cause it has taken into account a considerable number of real floods, trying to link reality to physical parameters, while most studies stay in the model world. To

- 50 increase the sample size, we also included flood events reported in the earlier years of the dataset (i.e. from the 1980s onwards). Regarding the missed events, we would like to emphasize that NatCatSERVICE database does not include a flood based on the hydrological definition (i.e. high water levels, peak discharges). Instead, an event enters the dataset when there is property damage and/or when there are people affected. Hence, the paper focuses only on these damaging events, which are usually the ones that humanitarian organizations are interested
- 55 in. However, by examining only the grid points where floods were reported and not all the grid points of sub-Saharan Africa, we have decreased the number of missed events. In the revised version, we explicitly mention these assumptions. Finally, when revising our paper, we have omitted analyses based on regional FPUs, since we feel that our dataset is too limited to identify enough data points per FPU.
- 60 COMMENT 6: It presents a very simplistic descriptive statistical evaluation.

RESPONSE 6: We agree that the original version of the paper presented a descriptive statistical rather than inferential statistical evaluation. In the revised version, we now perform statistical significance tests, which are carried out for; a) The SPEIs of flood and no-flood events b) The 7-day precipitation of flood and no-flood

- 5 events, c) the max 7-day precipitation during the flood month and a the 7-day precipitation of no-flood events. The no-flood cases considered refer, for each 'flooded cell', to the particular flood onset month of the no- flood years. Due to the very high number of no-flood events, the median value at all time scales is close to 0. The SPEI0-SPEI6 median values of floods are significantly higher, which is underpinned by the results of the z-test (p=0.05). More specifically, the median value of SPEI0, exhibits a value close to 1. This indicates that, as
- 10 expected, the wetness in the end of these months was high, demonstrating that SPEI0 could be used as a flood monitoring tool. On the (overlapping) seasonal time scales we see a positive relationship between reported floods and SPEI, which reduces while moving from SPEI1 to SPEI6 (0.5 to 0.1). In addition, we compare the maximum 7-day precipitation of each location during the no-flood years to the 7-day precipitation that preceded the flood events. For each flood, we standardized the 31 values over our 31 year of data (1 for the Flood (F) and
- 15 the 30 for No Floods (NF), with a mean of 0 and standard deviation of 1. This figure presents in boxplots the standardized 7-day precipitation (PRE7) of Flood (F) and No-Floods (NF) events. The results of the z-test showed that preceding PRE7 of floods did not exhibit any significant difference with that of no-floods (p= 0.1). This shows that although PRE7 that preceded the flood is high, it does not fully explains the flood generation. There were probably also similar magnitude events, in the same locations and during the same months that
- 20 floods were reported, that did not lead to a (reported) flood. Being aware of the dataset limitations (e.g. incapacity of reanalysis datasets to capture convective rainfall events, likely inaccurate onset date, etc.), the message that we want to convey is that since we observe a relation between seasonal SPEI and flooding and that this relation does not need a relation via weather-scale precipitation, implies that there probably is another factor that affects flooding on a seasonal scale prior to flood generation. One factor might be the soil saturation due to limited water storage capacity. This has been discussed in the revised version of the paper. Finally, we have
- 25 limited water storage capacity. This has been discussed in the revised version of the paper. Finally, we have omitted the original Figure 8 (SPEI per precipitation class) in the revised version, since conclusions based on this figure could not be supported with statistical significant differences.

COMMENT 7: It poorly acknowledges recent effort in seasonal forecasting (e.g. http://www.hydrol-earth- 30 systsci.net/special\_issue824.html).

RESPONSE 7: We thank Reviewer #2 for the link. In the revised version, we include some of these references (see 'Additional References').

35 Comments in the PDF

40

55

COMMENT 8: Consequently, they are more reliant on post-disaster response and preparedness activities, often assisted by international donors and humanitarian organizations. Consider here to have a look at Stephens et al. "Floodiness" approach. Stephens, E., J. J. Day, F. Pappenberger, and H. Cloke (2015), Precipitation and floodiness, Geophys. Res. Lett., 42, 10,316–10,323, doi:10.1002/2015GL066779.

RESPONSE 8: In the revised version, we have included this reference.

COMMENT 9: Here you might have a look at a recent HEPEX-HESS special issue on sub-seasonal to seasonal forecasting http://www.hydrol-earth-systsci.net/special\_issue824.html

RESPONSE 9: We have read some interesting and relevant papers of the HEPEX-HESS special issue and we have included some of them in the revised version (see 'Additional References').

50 COMMENT 10: The long-term ('seasonal-scale') wetness reflected in the SPEI for the preceding 1, 3 and 6 months, and (b) the short-term ('weather-scale') cumulative rainfall over the 7 days preceding the event. You could call this also long-memory and short memory disposition.

RESPONSE 10: Thank you for your remark. The text of the revised version has been changed significantly and we have taken this remark into consideration.

COMMENT 11: Figure 1: Schematic overview of the approach followed in this study. %False alarms seems a quite trivial metric for a sound assessment.

60 RESPONSE 11: We agree.

COMMENT 12: Munich Re NatCatSERVICE disaster database. Is there any cross validation of the accuracy/completeness of this data source?

- 5 RESPONSE 12: Unfortunately, we have not conducted any cross-validation of NatCatSERVICE database. The reason is that it is the only dataset at our disposal that provides details of reported flood events, such as coordinates, onset and end dates for the entire sub-Saharan Africa since 1980. We have also looked at other disaster datasets such as EM-DAT and DesInventar, but a systematic cross-validation was not possible as they did not have detailed geographical descriptions, and have only a limited number of reported floods.
- COMMENT 13: How you deal with the mismatch between the 0.5° and the 2.5° resolution? Wouldn't TRMM be an option to evaluate recent years?
- RESPONSE 13: We think that the datasets should be consistent in their spatial scale and therefore in the revised
   version, we have upscaled the 0.5° to 2.5° resolution, in order to take into account a larger flood affected area.
   TRMM provides observations only since 2000 and therefore we'd rather not use it for the sake of consistency throughout the paper.
- COMMENT 14: A statistical procedure was applied to fit the accumulated records to a log-logistic distribution with a mean of 0 and a standard deviation of 1. This should be shown or referenced

RESPONSE 14: We agree. In the methods sections of the revised manuscript we include the relevant references.

COMMENT 15: Above you speak about 7 days preceding the event, and here you speak about 7 days "lead time". In my understanding lead time is associated to forecasts

RESPONSE 15: That is correct. We agree that the 7 days preceding the event would rather be called "antecedent time" and not "lead time". We now only refer to 'lead time' when we talk about forecasts.

- 30 COMMENT 16: Figure 2: In this sketch I have the impression that you are dealing with a rather trivial problem, since 7 days prior to the flood peak you have already about 50% of the flood volume and 90% of the peak level. Please discuss. Minor: Add also SPEI0, Minor: Think about the word "lead time"
- RESPONSE 16: We thank reviewer #2 for this remark. We agree that the graph is confusing. The purpose of this
   graph is mainly to provide the reader a better understanding of the time points of SPEI and 7- day precipitation. The figure doesn't show any real flood event. We agree that usually a flood is defined with a significant discharge increase close the flood onset. In the revised manuscript, we replaced the graph with a more realistic one. Moreover, SPEI0 is shown on the map. Finally, we have substituted 'lead time' with 'antecedent time'.
- 40 COMMENT 17: How efficient is the flood reporting for each country?

50

55

RESPONSE 17: Unfortunately, to our knowledge, there is not any research that analyzes the efficiency of the flood reporting in each African country.

45 COMMENT 18: High correlation between flooding and wetter-than-average conditions. (trivial) RESPONSE 18 Indeed, we agree that this sentence does not give any important information to the results and we have removed it.

COMMENT 19: In how many cases was SPEI1> 1 but no flood was recorded?

RESPONSE 19: Taking into account all the 'flooded cells' and the months that in each one the flood was generated, we get 1731 cases with SPEI1>1 that no flood was recorded. This number accounts for 11.5% of all no-flood cases. In the revised version, we compare the percentages of flood and no-flood events that exceeded different thresholds and we quantify the elevated probability found for flood events.

- COMMENT 20: SPEI3 > 0.5 was slightly above average (52%), Is my understanding correct: If SPEI3 is > 0.5 then in about 50% of the cases you might expect a flood. Is this not very close to throwing a coin?
- RESPONSE 20: Our dataset consists of 501 floods over 31 years. Every flood is placed on a grid cell and
   therefore for each grid cell there is 1 flood and 30 no floods in the record. Therefore, there are 15030 cases of

no-floods. SPEI0>1 for 1666 no-flood cases (11%), SPEI1>1 for 1731 (11.5%) no- flood cases (11.5%), SPEI3>1 for 1571 no-flood cases (10.5%) and SPEI6>1 for 1454 no-flood cases (9.5%), while the corresponding percentages for floods is 41% (SPEI0), 27% (SPEI1), 21.5% (SPEI3), 16% (SPEI6). In the revised version, we present these increased probabilities. Moreover, in the revised manuscript, we also combine the seasonal SPEIs

with SPEI0 and 7-day precipitation. Following the same way of thinking, comparing the percentage of floods and no-floods that exhibited an SPEI3>0.5, we are arguing that it is twice more likely to have a flood in the location, where a flood was reported, when SPEI3>0.5.

COMMENT 21: Fig.7 Why not showing boxplots here?

RESPONSE 21: We have used boxplots in the revised version of our paper.

COMMENT 22: Limpopo basin - Check the references Seibert, M., Merz, B., and Apel, H.: Seasonal forecasting of hydrological drought in the Limpopo Basin: a comparison of statistical methods, Hydrol. Earth Syst. Sci., 21, 1611-1629, doi:10.5194/hess-21-1611-2017, 2017. Trambauer, P., Werner, M., Winsemius, H. C., Maskey, S., Dutra, E., and Uhlenbrook, S.: Hydrological drought forecasting and skill assessment for the Limpopo River basin, southern Africa, Hydrol. Earth Syst. Sci., 19, 1695-1711, doi:10.5194/hess-19-1695-2015, 2015. Dutra, E., Di Giuseppe, F., Wetterhall, F., and Pappenberger, F.: Seasonal forecasts of droughts in African basins using the Standardized Precipitation Index, Hydrol. Earth Syst. Sci., 17, 2359-2373, doi:10.5194/hess-17-2359-2013,

20 2013.

5

10

RESPONSE 22: We thank Reviewer #2 for these relevant and interesting articles. In the revised version, we have included include some of them (see 'Additional References').

25 COMMENT 23: (> 99th percentile) How many samples are in each 7-day precipitation category for the reported flood events?

RESPONSE 23: The samples in each 7- day category are: 0-33 percentile: 53 cases, 33-66 percentile: 119 cases, 66-100 percentile: 329 cases. However, since we did not find any statistically robust results, we have taken this part out.

1

30

COMMENT 24: Fig9. 5-colored boxplots welcome

RESPONSE 24: We have included boxplots in our figures.

COMMENT 25: Fig10 I try to understand, If SPEI1&0 are above 2, then in about 50% of the cases a flood occurred. Correct?

RESPONSE 25: Yes, that is correct. However, as explained in R.12, the no-flood cases are way more than the
 flood cases. In the revised version, we include figures, in which we show the elevated probability of having a flood when SPEI0 & SPEI1>0.

Additional References

45 In the revised document, we have included the following references;

Dutra, E., Di Giuseppe, F., Wetterhall, F., and Pappenberger, F.: Seasonal forecasts of droughts in African basins using the Standardized Precipitation Index, Hydrol. Earth Syst. Sci., 17, 2359-2373, doi:10.5194/hess-17-2359-2013, 2013

Stephens, E., J. J. Day, F. Pappenberger, and H. Cloke (2015), Precipitation and "n'Coodiness, ' Geophys. Res. Lett., 42, 10,316–10,323, doi:10.1002/2015GL066779

Coughlan de Perez, E., Stephens, E., Bischiniotis, K., van Aalst, M., van den Hurk, B., Mason, S., Nissan, H.,
 and Pappenberger, F.: Should seasonal rainfall forecasts be used for flood preparedness?, Hydrol. Earth Syst.
 Sci. Discuss., doi:10.5194/hess-2017-40, in review, 2017

Seibert, M., Merz, B., and Apel, H.: Seasonal forecasting of hydrological drought in the Limpopo Basin: a comparison of statistical methods, Hydrol. Earth Syst. Sci., 21, 1611-1629, doi:10.5194/hess-21-1611-2017, 2017

60

Zhang, Y., Moges, S., and Block, P.: Does objective cluster analysis serve as a useful precursor to seasonal precipitation prediction at local scale? Application to western Ethiopia, Hydrol. Earth Syst. Sci. Discuss., doi:10.5194/hess-2017-70, in review, 2017

- Tschoegl L., Below R. and GuhaâA× RSapir D. (2006). An Analytical review of selected data sets on × natural disasters and impacts. Paper prepared for the UNDP/CRED Workshop on Improving Compilation of Reliable Data on Disaster Occurrence and Impact. Bangkok, 2âA× R4 April 2008.
- 10 Below R, Wirtz, A. and Guha-Sapir D. (2009), Disaster category classification and peril terminology for operational purposes. Paper prepared fror CRED and Munich Re, October 2009.

Lorenz, C., and H. Kunstmann (2012), The Hydrological cycle in three state-of-the-art reanalyses: Intercomparison and performance analysis J. Hydrometeorol.,13(5), 1397–1420, doi:10.1175/JHM-D-11-088.1

- Herold, N., A. Behrangi, and L. V. Alexander (2017), Large uncertainties in observed daily precipitation extremes over land, J. Geophys. Res. Atmos., 122, 668–681, doi:10.1002/2016JD025842.
- Zhan, W., K. Guan, J. Sheffield, and E. F.Wood (2016), Depiction of drought oversub-Saharan Africa using
   reanalyses precipitation data sets J. Geophys. Res. Atmos., 121, 10,555–10,574, doi:10.1002/2016JD02485). 8

Zhang, Q., H. Körnich, and K. Holmgren (2013), How well do reanalyses represent the southern African precipitation?, Clim. Dyn., 40(3–4), 951–962, doi:10.1007/s00382-012-1423-z C

25

15

5

30

35

40

50

**The influence of antecedent conditions on flood risk in sub-Saharan Africa**

Konstantinos Bischiniotis1, Bart van den Hurk1,2, Brenden Jongman1,3, Erin Coughlan de Perez1,4,5, Ted Veldkamp1, Hans de Moel1, Jeroen Aerts1

(1) Institute for Environmental Studies, Vrije Universiteit Amsterdam, 1081 HV Amsterdam, The Netherlands
(2) Royal Netherlands Meteorological Institute (KNMI), De Bilt, 3731 GA, the Netherlands
(3) Global Facility for Disaster Reduction and Recovery (GFDRR), World Bank, Washington DC, USA
(4) International Research Institute for Climate and Society, Columbia University, Palisades, NY 10964, USA,

10 (5) Red Cross Red Crescent Climate Centre, The Hague, 2521 CV, the Netherlands

Correspondence to: Konstantinos Bischiniotis (kbischiniotis@gmail.com)

Abstract Most flood early warning systems have predominantly focused on forecasting floods with lead times of a hours or days. However, physical processes during longer time scales can also contribute to flood generation. In this study, we follow a pragmatic approach to analyse the hydro-meteorological pre-conditions of 501 historical

- 15 damaging floods over the period 1980 to 2010 in sub-Saharan Africa, These are separated into a) weather time scale (0-6 days) and b) seasonal time scale conditions (up to 6 months) before the event. The 7-day precipitation preceding a flood event (PRE7) and the Standardized Precipitation Evapotranspiration Index (SPEL) are analysed for the two time scale domains, respectively, Results indicate that high PRE7 does not always generate floods by itself. Seasonal SPEIs, which are not directly correlated with PRE7, exhibit positive (wet) values prior to most
- 20 flood events across different averaging times, indicating a relationship with flooding. The paper provides evidence that bringing together weather, and seasonal conditions can lead to improved flood risk preparedness.

**1 Introduction**

In recent decades, weather-related disasters have accounted for about 90% of all natural disasters (UNISDR, 2015a). There is an upward trend in disaster loss, which is driven by global climate change and the increasing concentration of populations and economic assets in flood-prone areas (Bouwer et al. 2007; Prenger-Berninghoff et al. 2014). Flooding affects millions of people across the globe each year. Between 1980 and 2012 the average annual reported losses and fatalities due to floods exceeded \$23 billion and 5,900 people, respectively (EM-DAT, 2012; Jongman et al., 2015).

Flood risk management has traditionally focused on long-term flood protection techniques such as levees and dams (Kellet
 and Caravani, 2013). Today, people employ complex combinations of flood risk strategies, ranging from technical flood protection measures to financial compensation mechanisms such as insurance, as well as nature-based solutions (Aerts et al., 2014). Lower-income countries often cannot afford and implement preventive measures, mainly due to the high investment costs (e.g. Douben, 2006). Consequently, they are more reliant on post-disaster response and preparedness activities, often assisted by international donors and humanitarian organizations.

35

25

5

The role of science in disaster risk reduction has been globally recognized in the Sendai Framework (UNISDR, 2015b). Preparedness activities and flood forecasting have received increasing attention and have led to new science-based early action systems (Coughlan de Perez et al., 2014). Weather forecasts, with typical lead times of some hours or days, have

| Formatted                                                                                                                                                                                                                                                                                                                                                                                                                                                                                                                                                                                                                                                                                                                                                                                                                                                                                                                                                                                                                                                                                                                                                                                                                                                                                                                                                                                                                                                                                                                                                                                                                                                                                                                                                                                                                                                                                                                                                                                                                                                                                                                                |                                                                                                                                                      |
|------------------------------------------------------------------------------------------------------------------------------------------------------------------------------------------------------------------------------------------------------------------------------------------------------------------------------------------------------------------------------------------------------------------------------------------------------------------------------------------------------------------------------------------------------------------------------------------------------------------------------------------------------------------------------------------------------------------------------------------------------------------------------------------------------------------------------------------------------------------------------------------------------------------------------------------------------------------------------------------------------------------------------------------------------------------------------------------------------------------------------------------------------------------------------------------------------------------------------------------------------------------------------------------------------------------------------------------------------------------------------------------------------------------------------------------------------------------------------------------------------------------------------------------------------------------------------------------------------------------------------------------------------------------------------------------------------------------------------------------------------------------------------------------------------------------------------------------------------------------------------------------------------------------------------------------------------------------------------------------------------------------------------------------------------------------------------------------------------------------------------------------|------------------------------------------------------------------------------------------------------------------------------------------------------|
|                                                                                                                                                                                                                                                                                                                                                                                                                                                                                                                                                                                                                                                                                                                                                                                                                                                                                                                                                                                                                                                                                                                                                                                                                                                                                                                                                                                                                                                                                                                                                                                                                                                                                                                                                                                                                                                                                                                                                                                                                                                                                                                                          | [1]                                                                                                                                                  |
| K Bischiniotis 22/7/2017 09:53                                                                                                                                                                                                                                                                                                                                                                                                                                                                                                                                                                                                                                                                                                                                                                                                                                                                                                                                                                                                                                                                                                                                                                                                                                                                                                                                                                                                                                                                                                                                                                                                                                                                                                                                                                                                                                                                                                                                                                                                                                                                                                           |                                                                                                                                                      |
| Formatted                                                                                                                                                                                                                                                                                                                                                                                                                                                                                                                                                                                                                                                                                                                                                                                                                                                                                                                                                                                                                                                                                                                                                                                                                                                                                                                                                                                                                                                                                                                                                                                                                                                                                                                                                                                                                                                                                                                                                                                                                                                                                                                                |                                                                                                                                                      |
| Formatted                                                                                                                                                                                                                                                                                                                                                                                                                                                                                                                                                                                                                                                                                                                                                                                                                                                                                                                                                                                                                                                                                                                                                                                                                                                                                                                                                                                                                                                                                                                                                                                                                                                                                                                                                                                                                                                                                                                                                                                                                                                                                                                                | [2]                                                                                                                                                  |
| K. Bischiniotis 22/7/2017 09:53                                                                                                                                                                                                                                                                                                                                                                                                                                                                                                                                                                                                                                                                                                                                                                                                                                                                                                                                                                                                                                                                                                                                                                                                                                                                                                                                                                                                                                                                                                                                                                                                                                                                                                                                                                                                                                                                                                                                                                                                                                                                                                          |                                                                                                                                                      |
| Formatted                                                                                                                                                                                                                                                                                                                                                                                                                                                                                                                                                                                                                                                                                                                                                                                                                                                                                                                                                                                                                                                                                                                                                                                                                                                                                                                                                                                                                                                                                                                                                                                                                                                                                                                                                                                                                                                                                                                                                                                                                                                                                                                                | [3]                                                                                                                                                  |
| K. Bischiniotis 22/7/2017 09:53                                                                                                                                                                                                                                                                                                                                                                                                                                                                                                                                                                                                                                                                                                                                                                                                                                                                                                                                                                                                                                                                                                                                                                                                                                                                                                                                                                                                                                                                                                                                                                                                                                                                                                                                                                                                                                                                                                                                                                                                                                                                                                          |                                                                                                                                                      |
| Formatted                                                                                                                                                                                                                                                                                                                                                                                                                                                                                                                                                                                                                                                                                                                                                                                                                                                                                                                                                                                                                                                                                                                                                                                                                                                                                                                                                                                                                                                                                                                                                                                                                                                                                                                                                                                                                                                                                                                                                                                                                                                                                                                                |                                                                                                                                                      |
| Tormatted                                                                                                                                                                                                                                                                                                                                                                                                                                                                                                                                                                                                                                                                                                                                                                                                                                                                                                                                                                                                                                                                                                                                                                                                                                                                                                                                                                                                                                                                                                                                                                                                                                                                                                                                                                                                                                                                                                                                                                                                                                                                                                                                |                                                                                                                                                      |
| K. Bischiniotis 22/7/2017 09:53                                                                                                                                                                                                                                                                                                                                                                                                                                                                                                                                                                                                                                                                                                                                                                                                                                                                                                                                                                                                                                                                                                                                                                                                                                                                                                                                                                                                                                                                                                                                                                                                                                                                                                                                                                                                                                                                                                                                                                                                                                                                                                          |                                                                                                                                                      |
| Formatted                                                                                                                                                                                                                                                                                                                                                                                                                                                                                                                                                                                                                                                                                                                                                                                                                                                                                                                                                                                                                                                                                                                                                                                                                                                                                                                                                                                                                                                                                                                                                                                                                                                                                                                                                                                                                                                                                                                                                                                                                                                                                                                                | [5]                                                                                                                                                  |
| K. Bischiniotis 22/7/2017 09:53                                                                                                                                                                                                                                                                                                                                                                                                                                                                                                                                                                                                                                                                                                                                                                                                                                                                                                                                                                                                                                                                                                                                                                                                                                                                                                                                                                                                                                                                                                                                                                                                                                                                                                                                                                                                                                                                                                                                                                                                                                                                                                          |                                                                                                                                                      |
| Formatted                                                                                                                                                                                                                                                                                                                                                                                                                                                                                                                                                                                                                                                                                                                                                                                                                                                                                                                                                                                                                                                                                                                                                                                                                                                                                                                                                                                                                                                                                                                                                                                                                                                                                                                                                                                                                                                                                                                                                                                                                                                                                                                                | [6]                                                                                                                                                  |
|                                                                                                                                                                                                                                                                                                                                                                                                                                                                                                                                                                                                                                                                                                                                                                                                                                                                                                                                                                                                                                                                                                                                                                                                                                                                                                                                                                                                                                                                                                                                                                                                                                                                                                                                                                                                                                                                                                                                                                                                                                                                                                                                          |                                                                                                                                                      |
| K. BISCHINIOUS 22/1/2017 09.55                                                                                                                                                                                                                                                                                                                                                                                                                                                                                                                                                                                                                                                                                                                                                                                                                                                                                                                                                                                                                                                                                                                                                                                                                                                                                                                                                                                                                                                                                                                                                                                                                                                                                                                                                                                                                                                                                                                                                                                                                                                                                                           |                                                                                                                                                      |
| Formatted                                                                                                                                                                                                                                                                                                                                                                                                                                                                                                                                                                                                                                                                                                                                                                                                                                                                                                                                                                                                                                                                                                                                                                                                                                                                                                                                                                                                                                                                                                                                                                                                                                                                                                                                                                                                                                                                                                                                                                                                                                                                                                                                | [8]                                                                                                                                                  |
| K. Bischiniotis 22/7/2017 09:53                                                                                                                                                                                                                                                                                                                                                                                                                                                                                                                                                                                                                                                                                                                                                                                                                                                                                                                                                                                                                                                                                                                                                                                                                                                                                                                                                                                                                                                                                                                                                                                                                                                                                                                                                                                                                                                                                                                                                                                                                                                                                                          |                                                                                                                                                      |
| Formatted                                                                                                                                                                                                                                                                                                                                                                                                                                                                                                                                                                                                                                                                                                                                                                                                                                                                                                                                                                                                                                                                                                                                                                                                                                                                                                                                                                                                                                                                                                                                                                                                                                                                                                                                                                                                                                                                                                                                                                                                                                                                                                                                | ([7])                                                                                                                                                |
| K Bischiniotis 22/7/2017 09:53                                                                                                                                                                                                                                                                                                                                                                                                                                                                                                                                                                                                                                                                                                                                                                                                                                                                                                                                                                                                                                                                                                                                                                                                                                                                                                                                                                                                                                                                                                                                                                                                                                                                                                                                                                                                                                                                                                                                                                                                                                                                                                           |                                                                                                                                                      |
| Related                                                                                                                                                                                                                                                                                                                                                                                                                                                                                                                                                                                                                                                                                                                                                                                                                                                                                                                                                                                                                                                                                                                                                                                                                                                                                                                                                                                                                                                                                                                                                                                                                                                                                                                                                                                                                                                                                                                                                                                                                                                                                                                                  |                                                                                                                                                      |
| Deleted: -seasonal-                                                                                                                                                                                                                                                                                                                                                                                                                                                                                                                                                                                                                                                                                                                                                                                                                                                                                                                                                                                                                                                                                                                                                                                                                                                                                                                                                                                                                                                                                                                                                                                                                                                                                                                                                                                                                                                                                                                                                                                                                                                                                                                      |                                                                                                                                                      |
| K. Bischiniotis 22/7/2017 09:53                                                                                                                                                                                                                                                                                                                                                                                                                                                                                                                                                                                                                                                                                                                                                                                                                                                                                                                                                                                                                                                                                                                                                                                                                                                                                                                                                                                                                                                                                                                                                                                                                                                                                                                                                                                                                                                                                                                                                                                                                                                                                                          |                                                                                                                                                      |
| Formatted                                                                                                                                                                                                                                                                                                                                                                                                                                                                                                                                                                                                                                                                                                                                                                                                                                                                                                                                                                                                                                                                                                                                                                                                                                                                                                                                                                                                                                                                                                                                                                                                                                                                                                                                                                                                                                                                                                                                                                                                                                                                                                                                | [9]                                                                                                                                                  |
| K Bischiniotis 22/7/2017 09:53                                                                                                                                                                                                                                                                                                                                                                                                                                                                                                                                                                                                                                                                                                                                                                                                                                                                                                                                                                                                                                                                                                                                                                                                                                                                                                                                                                                                                                                                                                                                                                                                                                                                                                                                                                                                                                                                                                                                                                                                                                                                                                           |                                                                                                                                                      |
| Deleted: flood events                                                                                                                                                                                                                                                                                                                                                                                                                                                                                                                                                                                                                                                                                                                                                                                                                                                                                                                                                                                                                                                                                                                                                                                                                                                                                                                                                                                                                                                                                                                                                                                                                                                                                                                                                                                                                                                                                                                                                                                                                                                                                                                    |                                                                                                                                                      |
|                                                                                                                                                                                                                                                                                                                                                                                                                                                                                                                                                                                                                                                                                                                                                                                                                                                                                                                                                                                                                                                                                                                                                                                                                                                                                                                                                                                                                                                                                                                                                                                                                                                                                                                                                                                                                                                                                                                                                                                                                                                                                                                                          |                                                                                                                                                      |
| K. Bischiniotis 22/7/2017 09:53                                                                                                                                                                                                                                                                                                                                                                                                                                                                                                                                                                                                                                                                                                                                                                                                                                                                                                                                                                                                                                                                                                                                                                                                                                                                                                                                                                                                                                                                                                                                                                                                                                                                                                                                                                                                                                                                                                                                                                                                                                                                                                          |                                                                                                                                                      |
| Formatted                                                                                                                                                                                                                                                                                                                                                                                                                                                                                                                                                                                                                                                                                                                                                                                                                                                                                                                                                                                                                                                                                                                                                                                                                                                                                                                                                                                                                                                                                                                                                                                                                                                                                                                                                                                                                                                                                                                                                                                                                                                                                                                                | [10]                                                                                                                                                 |
| K. Bischiniotis 22/7/2017 09:53                                                                                                                                                                                                                                                                                                                                                                                                                                                                                                                                                                                                                                                                                                                                                                                                                                                                                                                                                                                                                                                                                                                                                                                                                                                                                                                                                                                                                                                                                                                                                                                                                                                                                                                                                                                                                                                                                                                                                                                                                                                                                                          |                                                                                                                                                      |
| Deleted: are analysed.                                                                                                                                                                                                                                                                                                                                                                                                                                                                                                                                                                                                                                                                                                                                                                                                                                                                                                                                                                                                                                                                                                                                                                                                                                                                                                                                                                                                                                                                                                                                                                                                                                                                                                                                                                                                                                                                                                                                                                                                                                                                                                                   |                                                                                                                                                      |
| K Bischipiotic 22/7/2017 00-52                                                                                                                                                                                                                                                                                                                                                                                                                                                                                                                                                                                                                                                                                                                                                                                                                                                                                                                                                                                                                                                                                                                                                                                                                                                                                                                                                                                                                                                                                                                                                                                                                                                                                                                                                                                                                                                                                                                                                                                                                                                                                                           |                                                                                                                                                      |
| R. BISCHIHOUS 22/7/2017 09:53                                                                                                                                                                                                                                                                                                                                                                                                                                                                                                                                                                                                                                                                                                                                                                                                                                                                                                                                                                                                                                                                                                                                                                                                                                                                                                                                                                                                                                                                                                                                                                                                                                                                                                                                                                                                                                                                                                                                                                                                                                                                                                            |                                                                                                                                                      |
| Formatted                                                                                                                                                                                                                                                                                                                                                                                                                                                                                                                                                                                                                                                                                                                                                                                                                                                                                                                                                                                                                                                                                                                                                                                                                                                                                                                                                                                                                                                                                                                                                                                                                                                                                                                                                                                                                                                                                                                                                                                                                                                                                                                                | [11]                                                                                                                                                 |
| K. Bischiniotis 22/7/2017 09:53                                                                                                                                                                                                                                                                                                                                                                                                                                                                                                                                                                                                                                                                                                                                                                                                                                                                                                                                                                                                                                                                                                                                                                                                                                                                                                                                                                                                                                                                                                                                                                                                                                                                                                                                                                                                                                                                                                                                                                                                                                                                                                          |                                                                                                                                                      |
| Deleted: a short-term '                                                                                                                                                                                                                                                                                                                                                                                                                                                                                                                                                                                                                                                                                                                                                                                                                                                                                                                                                                                                                                                                                                                                                                                                                                                                                                                                                                                                                                                                                                                                                                                                                                                                                                                                                                                                                                                                                                                                                                                                                                                                                                                  |                                                                                                                                                      |
| K Bischiniotis 22/7/2017 09:53                                                                                                                                                                                                                                                                                                                                                                                                                                                                                                                                                                                                                                                                                                                                                                                                                                                                                                                                                                                                                                                                                                                                                                                                                                                                                                                                                                                                                                                                                                                                                                                                                                                                                                                                                                                                                                                                                                                                                                                                                                                                                                           |                                                                                                                                                      |
| Formattad                                                                                                                                                                                                                                                                                                                                                                                                                                                                                                                                                                                                                                                                                                                                                                                                                                                                                                                                                                                                                                                                                                                                                                                                                                                                                                                                                                                                                                                                                                                                                                                                                                                                                                                                                                                                                                                                                                                                                                                                                                                                                                                                |                                                                                                                                                      |
| Tormatted                                                                                                                                                                                                                                                                                                                                                                                                                                                                                                                                                                                                                                                                                                                                                                                                                                                                                                                                                                                                                                                                                                                                                                                                                                                                                                                                                                                                                                                                                                                                                                                                                                                                                                                                                                                                                                                                                                                                                                                                                                                                                                                                |                                                                                                                                                      |
| K. Bischiniotis 22/7/2017 09:53                                                                                                                                                                                                                                                                                                                                                                                                                                                                                                                                                                                                                                                                                                                                                                                                                                                                                                                                                                                                                                                                                                                                                                                                                                                                                                                                                                                                                                                                                                                                                                                                                                                                                                                                                                                                                                                                                                                                                                                                                                                                                                          |                                                                                                                                                      |
| Deleted: scale' period                                                                                                                                                                                                                                                                                                                                                                                                                                                                                                                                                                                                                                                                                                                                                                                                                                                                                                                                                                                                                                                                                                                                                                                                                                                                                                                                                                                                                                                                                                                                                                                                                                                                                                                                                                                                                                                                                                                                                                                                                                                                                                                   |                                                                                                                                                      |
| K. Bischiniotis 22/7/2017 09:53                                                                                                                                                                                                                                                                                                                                                                                                                                                                                                                                                                                                                                                                                                                                                                                                                                                                                                                                                                                                                                                                                                                                                                                                                                                                                                                                                                                                                                                                                                                                                                                                                                                                                                                                                                                                                                                                                                                                                                                                                                                                                                          |                                                                                                                                                      |
| Formatted                                                                                                                                                                                                                                                                                                                                                                                                                                                                                                                                                                                                                                                                                                                                                                                                                                                                                                                                                                                                                                                                                                                                                                                                                                                                                                                                                                                                                                                                                                                                                                                                                                                                                                                                                                                                                                                                                                                                                                                                                                                                                                                                | [13]                                                                                                                                                 |
| K. Bischiniotis 22/7/2017 09:53                                                                                                                                                                                                                                                                                                                                                                                                                                                                                                                                                                                                                                                                                                                                                                                                                                                                                                                                                                                                                                                                                                                                                                                                                                                                                                                                                                                                                                                                                                                                                                                                                                                                                                                                                                                                                                                                                                                                                                                                                                                                                                          |                                                                                                                                                      |
| Deleted: 7                                                                                                                                                                                                                                                                                                                                                                                                                                                                                                                                                                                                                                                                                                                                                                                                                                                                                                                                                                                                                                                                                                                                                                                                                                                                                                                                                                                                                                                                                                                                                                                                                                                                                                                                                                                                                                                                                                                                                                                                                                                                                                                               |                                                                                                                                                      |
| K Bischiniotis 22/7/2017 00:53                                                                                                                                                                                                                                                                                                                                                                                                                                                                                                                                                                                                                                                                                                                                                                                                                                                                                                                                                                                                                                                                                                                                                                                                                                                                                                                                                                                                                                                                                                                                                                                                                                                                                                                                                                                                                                                                                                                                                                                                                                                                                                           | $ \rightarrow $                                                                                                                                      |
| Formattad                                                                                                                                                                                                                                                                                                                                                                                                                                                                                                                                                                                                                                                                                                                                                                                                                                                                                                                                                                                                                                                                                                                                                                                                                                                                                                                                                                                                                                                                                                                                                                                                                                                                                                                                                                                                                                                                                                                                                                                                                                                                                                                                |                                                                                                                                                      |
| Formatteu                                                                                                                                                                                                                                                                                                                                                                                                                                                                                                                                                                                                                                                                                                                                                                                                                                                                                                                                                                                                                                                                                                                                                                                                                                                                                                                                                                                                                                                                                                                                                                                                                                                                                                                                                                                                                                                                                                                                                                                                                                                                                                                                | C                                                                                                                                                    |
|                                                                                                                                                                                                                                                                                                                                                                                                                                                                                                                                                                                                                                                                                                                                                                                                                                                                                                                                                                                                                                                                                                                                                                                                                                                                                                                                                                                                                                                                                                                                                                                                                                                                                                                                                                                                                                                                                                                                                                                                                                                                                                                                          | [14] )                                                                                                                                               |
| K. Bischiniotis 22/7/2017 09:53                                                                                                                                                                                                                                                                                                                                                                                                                                                                                                                                                                                                                                                                                                                                                                                                                                                                                                                                                                                                                                                                                                                                                                                                                                                                                                                                                                                                                                                                                                                                                                                                                                                                                                                                                                                                                                                                                                                                                                                                                                                                                                          | [14]                                                                                                                                                 |
| K. Bischiniotis 22/7/2017 09:53
Deleted: a long-term '                                                                                                                                                                                                                                                                                                                                                                                                                                                                                                                                                                                                                                                                                                                                                                                                                                                                                                                                                                                                                                                                                                                                                                                                                                                                                                                                                                                                                                                                                                                                                                                                                                                                                                                                                                                                                                                                                                                                                                                                                                                                                | [14]                                                                                                                                                 |
| K. Bischiniotis 22/7/2017 09:53
K. Bischiniotis 22/7/2017 09:53                                                                                                                                                                                                                                                                                                                                                                                                                                                                                                                                                                                                                                                                                                                                                                                                                                                                                                                                                                                                                                                                                                                                                                                                                                                                                                                                                                                                                                                                                                                                                                                                                                                                                                                                                                                                                                                                                                                                                                                                                                             | [14]                                                                                                                                                 |
| K. Bischiniotis 22/7/2017 09:53
K. Bischiniotis 22/7/2017 09:53
Formatted                                                                                                                                                                                                                                                                                                                                                                                                                                                                                                                                                                                                                                                                                                                                                                                                                                                                                                                                                                                                                                                                                                                                                                                                                                                                                                                                                                                                                                                                                                                                                                                                                                                                                                                                                                                                                                                                                                                                                                                                                                | [14]                                                                                                                                                 |
| K. Bischiniotis 22/7/2017 09:53
K. Bischiniotis 22/7/2017 09:53
Formatted
K. Bischiniotis 22/7/2017 09:53                                                                                                                                                                                                                                                                                                                                                                                                                                                                                                                                                                                                                                                                                                                                                                                                                                                                                                                                                                                                                                                                                                                                                                                                                                                                                                                                                                                                                                                                                                                                                                                                                                                                                                                                                                                                                                                                                                                                                                                             | [14] )
[15]                                                                                                                                       |
| K. Bischiniotis 22/7/2017 09:53
K. Bischiniotis 22/7/2017 09:53
Formatted
K. Bischiniotis 22/7/2017 09:53
Deleted: scale' period                                                                                                                                                                                                                                                                                                                                                                                                                                                                                                                                                                                                                                                                                                                                                                                                                                                                                                                                                                                                                                                                                                                                                                                                                                                                                                                                                                                                                                                                                                                                                                                                                                                                                                                                                                                                                                                                                                                                                                   | [14] )
[15]                                                                                                                                       |
| K. Bischiniotis 22/7/2017 09:53
K. Bischiniotis 22/7/2017 09:53
Formatted
K. Bischiniotis 22/7/2017 09:53
Deleted: -scale' period                                                                                                                                                                                                                                                                                                                                                                                                                                                                                                                                                                                                                                                                                                                                                                                                                                                                                                                                                                                                                                                                                                                                                                                                                                                                                                                                                                                                                                                                                                                                                                                                                                                                                                                                                                                                                                                                                                                                             | [14]                                                                                                                                                 |
| K. Bischiniotis 22/7/2017 09:53
K. Bischiniotis 22/7/2017 09:53
Formatted
K. Bischiniotis 22/7/2017 09:53
K. Bischiniotis 22/7/2017 09:53                                                                                                                                                                                                                                                                                                                                                                                                                                                                                                                                                                                                                                                                                                                                                                                                                                                                                                                                                                                                                                                                                                                                                                                                                                                                                                                                                                                                                                                                                                                                                                                                                                                                                                                                                                                                                                                                                                                               | [14]                                                                                                                                                 |
| K. Bischiniotis 22/7/2017 09:53
K. Bischiniotis 22/7/2017 09:53
Formatted
K. Bischiniotis 22/7/2017 09:53
K. Bischiniotis 22/7/2017 09:53
Formatted                                                                                                                                                                                                                                                                                                                                                                                                                                                                                                                                                                                                                                                                                                                                                                                                                                                                                                                                                                                                                                                                                                                                                                                                                                                                                                                                                                                                                                                                                                                                                                                                                                                                                                                                                                                                                                                                                                                  | [14]                                                                                                                                                 |
| K. Bischiniotis 22/7/2017 09:53
K. Bischiniotis 22/7/2017 09:53
Formatted
K. Bischiniotis 22/7/2017 09:53
K. Bischiniotis 22/7/2017 09:53
Formatted
K. Bischiniotis 22/7/2017 09:53                                                                                                                                                                                                                                                                                                                                                                                                                                                                                                                                                                                                                                                                                                                                                                                                                                                                                                                                                                                                                                                                                                                                                                                                                                                                                                                                                                                                                                                                                                                                                                                                                                                                                                                                                                                                                                                                               | [14]                                                                                                                                                 |
| K. Bischiniotis 22/7/2017 09:53
K. Bischiniotis 22/7/2017 09:53
Formatted
K. Bischiniotis 22/7/2017 09:53
K. Bischiniotis 22/7/2017 09:53
Formatted
K. Bischiniotis 22/7/2017 09:53
Deleted: flood                                                                                                                                                                                                                                                                                                                                                                                                                                                                                                                                                                                                                                                                                                                                                                                                                                                                                                                                                                                                                                                                                                                                                                                                                                                                                                                                                                                                                                                                                                                                                                                                                                                                                                                                                                                                                                                             | [14]                                                                                                                                                 |
|  <li>K. Bischiniotis 22/7/2017 09:53</li> <li>Deleted: a long-term '</li> <li>K. Bischiniotis 22/7/2017 09:53</li> <li>Formatted</li> <li>K. Bischiniotis 22/7/2017 09:53</li> <li>Deleted: -scale' period</li> <li>K. Bischiniotis 22/7/2017 09:53</li> <li>Formatted</li> <li>K. Bischiniotis 22/7/2017 09:53</li> <li>Deleted: flood</li> <li>K. Bischiniotis 22/7/2017 09:53</li>                                                                                                                                                                                                                                                                                                                                                                                                                                                                                                                                                                                                                                                                                                                                                                                                                                                                                                                                                                                                                                                                                                                                                                                                                                                                                                                                                                                                                                                                                                                                                                                                                                                                                                                                           | [14])
[15])
[16])                                                                                                                              |
| K. Bischiniotis 22/7/2017 09:53
K. Bischiniotis 22/7/2017 09:53
Formatted
K. Bischiniotis 22/7/2017 09:53
K. Bischiniotis 22/7/2017 09:53
Formatted
K. Bischiniotis 22/7/2017 09:53
K. Bischiniotis 22/7/2017 09:53
Ecomatted                                                                                                                                                                                                                                                                                                                                                                                                                                                                                                                                                                                                                                                                                                                                                                                                                                                                                                                                                                                                                                                                                                                                                                                                                                                                                                                                                                                                                                                                                                                                                                                                                                                                                                                                                                                                             |                                                                                                                                                      |
| K. Bischiniotis 22/7/2017 09:53
K. Bischiniotis 22/7/2017 09:53
Formatted
K. Bischiniotis 22/7/2017 09:53
K. Bischiniotis 22/7/2017 09:53
Formatted
K. Bischiniotis 22/7/2017 09:53
K. Bischiniotis 22/7/2017 09:53
Formatted                                                                                                                                                                                                                                                                                                                                                                                                                                                                                                                                                                                                                                                                                                                                                                                                                                                                                                                                                                                                                                                                                                                                                                                                                                                                                                                                                                                                                                                                                                                                                                                                                                                                                                                                                                                                             | [14]
[15]
[16]                                                                                                                                 |
| K. Bischiniotis 22/7/2017 09:53
K. Bischiniotis 22/7/2017 09:53
Formatted
K. Bischiniotis 22/7/2017 09:53
K. Bischiniotis 22/7/2017 09:53
Formatted
K. Bischiniotis 22/7/2017 09:53
K. Bischiniotis 22/7/2017 09:53
Formatted
K. Bischiniotis 22/7/2017 09:53                                                                                                                                                                                                                                                                                                                                                                                                                                                                                                                                                                                                                                                                                                                                                                                                                                                                                                                                                                                                                                                                                                                                                                                                                                                                                                                                                                                                                                                                                                                                                                                                                                                                                                                                                                          | [14]
[15]
[16]
[17]                                                                                                                         |
| K. Bischiniotis 22/7/2017 09:53
K. Bischiniotis 22/7/2017 09:53
Formatted
K. Bischiniotis 22/7/2017 09:53
K. Bischiniotis 22/7/2017 09:53
Formatted
K. Bischiniotis 22/7/2017 09:53
K. Bischiniotis 22/7/2017 09:53
Formatted
K. Bischiniotis 22/7/2017 09:53
Deleted: Total                                                                                                                                                                                                                                                                                                                                                                                                                                                                                                                                                                                                                                                                                                                                                                                                                                                                                                                                                                                                                                                                                                                                                                                                                                                                                                                                                                                                                                                                                                                                                                                                                                                                                                                                                        | [14]
[15]
[16]
[17]                                                                                                                         |
| K. Bischiniotis 22/7/2017 09:53
K. Bischiniotis 22/7/2017 09:53
Formatted
K. Bischiniotis 22/7/2017 09:53
K. Bischiniotis 22/7/2017 09:53
Formatted
K. Bischiniotis 22/7/2017 09:53
K. Bischiniotis 22/7/2017 09:53
Formatted
K. Bischiniotis 22/7/2017 09:53
K. Bischiniotis 22/7/2017 09:53                                                                                                                                                                                                                                                                                                                                                                                                                                                                                                                                                                                                                                                                                                                                                                                                                                                                                                                                                                                                                                                                                                                                                                                                                                                                                                                                                                                                                                                                                                                                                                                                                                                                                                                     | [14]
                                                                                                                                             |
| K. Bischiniotis 22/7/2017 09:53
K. Bischiniotis 22/7/2017 09:53
Formatted
K. Bischiniotis 22/7/2017 09:53
K. Bischiniotis 22/7/2017 09:53
Formatted
K. Bischiniotis 22/7/2017 09:53
K. Bischiniotis 22/7/2017 09:53
K. Bischiniotis 22/7/2017 09:53
K. Bischiniotis 22/7/2017 09:53
Formatted                                                                                                                                                                                                                                                                                                                                                                                                                                                                                                                                                                                                                                                                                                                                                                                                                                                                                                                                                                                                                                                                                                                                                                                                                                                                                                                                                                                                                                                                                                                                                                                                                                                                                                   | [14]
[15]
[16]
[17]                                                                                                                         |
|  <li>K. Bischiniotis 22/7/2017 09:53</li> <li>Deleted: a long-term '</li> <li>K. Bischiniotis 22/7/2017 09:53</li> <li>Formatted</li> <li>K. Bischiniotis 22/7/2017 09:53</li> <li>Deleted: -scale' period</li> <li>K. Bischiniotis 22/7/2017 09:53</li> <li>Formatted</li> <li>K. Bischiniotis 22/7/2017 09:53</li> <li>Deleted: flood</li> <li>K. Bischiniotis 22/7/2017 09:53</li> <li>Formatted</li> <li>K. Bischiniotis 22/7/2017 09:53</li> <li>Deleted: Total</li> <li>K. Bischiniotis 22/7/2017 09:53</li> <li>Formatted</li> <li>K. Bischiniotis 22/7/2017 09:53</li> <li>Deleted: Total</li> <li>K. Bischiniotis 22/7/2017 09:53</li> <li>Formatted</li> <li>K. Bischiniotis 22/7/2017 09:53</li>                                                                                                                                                                                                                                                                                                                                                                                                                                                                                                                                                                                                                                                                                                                                                                                                                                                                                                                                                                                                                                                                                                                                                                                                                                                                                                                                                                                                                     | [14]                                                                                                                                                 |
|  <li>K. Bischiniotis 22/7/2017 09:53</li> <li>Deleted: a long-term '</li> <li>K. Bischiniotis 22/7/2017 09:53</li> <li>Formatted</li> <li>K. Bischiniotis 22/7/2017 09:53</li> <li>Deleted: -scale' period</li> <li>K. Bischiniotis 22/7/2017 09:53</li> <li>Formatted</li> <li>K. Bischiniotis 22/7/2017 09:53</li> <li>Deleted: flood</li> <li>K. Bischiniotis 22/7/2017 09:53</li> <li>Formatted</li> <li>K. Bischiniotis 22/7/2017 09:53</li> <li>Deleted: Total</li> <li>K. Bischiniotis 22/7/2017 09:53</li> <li>Formatted</li> <li>K. Bischiniotis 22/7/2017 09:53</li> <li>Deleted: Total</li> <li>K. Bischiniotis 22/7/2017 09:53</li> <li>Deleted: is used to acquise upstate and the scale</li>                                                                                                                                                                                                                                                                                                                                                                                                                                                                                                                                                                                                                                                                                                                                                                                                                                                                                                                                                                                                                                                                                                                                                                                                                                                                                                                                                                                                                      | [14]                                                                                                                                                 |
|  <li>K. Bischiniotis 22/7/2017 09:53</li> <li>Deleted: a long-term '</li> <li>K. Bischiniotis 22/7/2017 09:53</li> <li>Formatted</li> <li>K. Bischiniotis 22/7/2017 09:53</li> <li>Deleted: -scale' period</li> <li>K. Bischiniotis 22/7/2017 09:53</li> <li>Formatted</li> <li>K. Bischiniotis 22/7/2017 09:53</li> <li>Deleted: flood</li> <li>K. Bischiniotis 22/7/2017 09:53</li> <li>Deleted: Total</li> <li>K. Bischiniotis 22/7/2017 09:53</li> <li>Deleted: Total</li> <li>K. Bischiniotis 22/7/2017 09:53</li> <li>Deleted: is used to evaluate weather-scal</li> <li>K. Dischiniotis 22/7/2017 09:53</li>                                                                                                                                                                                                                                                                                                                                                                                                                                                                                                                                                                                                                                                                                                                                                                                                                                                                                                                                                                                                                                                                                                                                                                                                                                                                                                                                                                                                                                                                                                             | [14]                                                                                                                                                 |
| K. Bischiniotis 22/7/2017 09:53
K. Bischiniotis 22/7/2017 09:53
Formatted
K. Bischiniotis 22/7/2017 09:53
K. Bischiniotis 22/7/2017 09:53
Formatted
K. Bischiniotis 22/7/2017 09:53
K. Bischiniotis 22/7/2017 09:53
K. Bischiniotis 22/7/2017 09:53
Formatted
K. Bischiniotis 22/7/2017 09:53
K. Bischiniotis 22/7/2017 09:53
K. Bischiniotis 22/7/2017 09:53                                                                                                                                                                                                                                                                                                                                                                                                                                                                                                                                                                                                                                                                                                                                                                                                                                                                                                                                                                                                                                                                                                                                                                                                                                                                                                                                                                                                                                                                                                                                                     | [14]
[15]
[15]
[17]
                                                                                                                     |
|  <li>K. Bischiniotis 22/7/2017 09:53</li> <li>Deleted: a long-term '</li> <li>K. Bischiniotis 22/7/2017 09:53</li> <li>Formatted</li> <li>K. Bischiniotis 22/7/2017 09:53</li> <li>Deleted: -scale' period</li> <li>K. Bischiniotis 22/7/2017 09:53</li> <li>Formatted</li> <li>K. Bischiniotis 22/7/2017 09:53</li> <li>Deleted: flood</li> <li>K. Bischiniotis 22/7/2017 0

---

## Author Response (AR2)

*List of changes compared to the previous manuscript*

- Changes in the text regarding

  - The methodology
  - The quality of the data and its shortcomings
  - The potential implementation of the method in early warning schemes
  - The interpretation of the results

- Methodology

  - Define the 7-day precipitation based on SPI calculation (in order to address Reviewer #1 comment #3a)
  - Calculate the Risk Ratio and its confidence intervals (in order to address Reviewer #1 comment #3c and the Editor's comment #3)
15
  - Change Figures 7, 8, 9
  - Create Figures for the supplementary material (S.1-S.16)

- Extra references
20
  - Other studies that have used Munich Re database in their analysis
  - Methodology of estimating the Risk Ratio and its confidence intervals
  - Studies that have used the Risk Ratio

40

45

50

55

**Response to reviewers and to the editor**

**Title:** The influence of antecedent conditions on flood risk in sub-Saharan Africa

**Authors:** Konstantinos Bischiniotis, Bart van den Hurk, Brenden Jongman, Erin Coughlan de Perez, Ted Veldkamp, Hans de Moel, Jeroen Aerts
* * *
**General response**

We thank the two reviewers and the editor for the time taken to review and process our manuscript. In response to their comments, we have further revised our manuscript, and we have made additions and changes as outlined in this document. In the following sections, we respond to each of the reviewers' remarks or questions. Our responses are colored in blue.
* * *
**Editor:**

Thanks a lot for your revised manuscript. The 2 referees of the original version have looked at the revisions and have come to conflicting recommendations. Reviewer #2 is happy with your revisions, whereas Reviewer #1 recommends to reject the paper, although he/she acknowledges that the manuscript has improved compared to the original version. Given this situation, I have had a close look at the manuscript and decided on major revisions. I feel that, on the one hand, the concerns of Reviewer #1 are valid, but on the other hand, I think that the paper can hopefully be published after further revisions. I comment the 3 major concerns that reviewer #1 has raised:

We thank the editor for his detailed comments and for the valuable suggestions to further improve manuscript. We also think that Reviewer #1 comments are valid and we have revised the manuscript so as to address them as good as possible.

1)    I agree with the reviewer that there is little information on the climatic and hydrological setting, and the Köppen classification used does not seem to bring further insight to the topic. It would be great if the variation of flood generation across your study area could be discussed – but on the other hand I would understand that, given the coarseness of the data and the study design (pooling of all data across the study area), the actual benefit of such a discussion might be small and might not be worth a large effort.

We agree with the editor and with Reviewer #1 that not much information that could further bring insight to the topic can be derived from the Köppen classification. When classifying the reported flood events per climatological areas, our sample size per area becomes even smaller, as the editor suggested. For example, when using Level 9 Köppen classification, the average sample size per area is 60 floods. This, in combination with the fact that local conditions play a substantial role in flood generation, are likely he main reasons that no robust conclusions can be drawn. We have addressed these issues in the revised paper. We have not drawn any conclusions based on this, and simply present the locations, where the floods were reported.

2)    Concerning the 'data quality' issue, the reviewer might be right that the data quality is so low, and the confounding factors so various, that a statistical analysis always contains weak signals and much noise. My recommendation is that you check whether all your statements on the use of the results and the potential for improving/implementing early warning and preparedness schemes are valid. I feel that there would be a very long way to go, and in the end it might not be possible to gain much along this path.

We thank the editor for his comment and recommendation. Throughout the text, we have explicitly mentioned that the analysis is conducted using floods from Munich Re disaster database, acknowledging its shortcomings. Nevertheless, we argue that Munich Re is the most detailed and most complete disaster database currently available at the scales and regions we apply our method. For example, according to Vries, M. (2017), Munich Re has reported the most flood events in Tanzania, Malaysia and Ireland, including all the events that have been reported in FloodList, Flood Observatory and EM_DAT. Moreover, Munich Re database has also been applied in several scientific studies (e.g. Hoeppe, 2016; Jongman et al., 2014a) Based on the editor's suggestion, we have checked the statements we have made regarding the quality of the data, how this is related with improving/implementing early warning systems and why this study is useful to the humanitarian sector.

3)    I feel that the concerns of the referee on 'statistical methods' should be taken very seriously. To use simple, pragmatic methods is not a problem in case they are appropriate and robust, but I ask you to really consider his/her questions about the methodology, in particular his/her point about the robustness of the results. For example, would it be possible to provide uncertainty intervals for the 'relative odds of floods vs NF'? Figures 7 and 8 show a

strange behaviour for large SPEI values which somehow undermines the confidence in the analysis. Further, the statement (based on Fig. 5b) that max-7-day-precip is significantly different between floods and non-floods should be carefully interpreted: The difference might be statistically significant (partly because statistical tests tend to find significance when the sample size gets large), but maybe not relevant given the variability in both samples.

We thank the editor for his comments and suggestions. In the new revised version, we have taken into account Reviewer #1 comments, and we have further extended the statistical analysis to address his/her concerns.

First, we have altered the name 'relative odds' to 'risk ratio' (RR) as it is a term that is frequently used in literature, especially in medical and epidemiology studies (e.g. Katz, 2006; Shrier and Steele, 2006; Zhang, 1998).
Subsequently, following the methodology of Moris and Gardner (1988), we calculated the confidence intervals of this risk ratio. The principle in this methodology is that although the sample does not follow a normal distribution, its natural logarithm is approximately normally distributed to produce the 95% confidence intervals. Therefore, first, a confidence interval is generated for $\log_e(RR)$ and subsequently, the antilog of the upper and lower limits of the confidence interval for $\log_e(RR)$ are computed to give the upper and lower limits of the confidence interval for the RR ). In case, the upper limit is above 1 and the lower limit below 1, the risk ratio is not statistically significant. An analytical description is given in Section 2 of the revised manuscript.
For each case of Figures 1, 2, and 3 (see the end of this document), which correspond to Figures 7, 8, and 9 of the original manuscript, we have created a graph that shows the confidence intervals (Figures 4-19). The fact that high SPEI values show a strange behaviour is a consequence of the large confidence intervals that the small number of floods and no-floods produce. Note that for SPEI0, which includes the period in which the reported floods occurred, an expected monotonous increase of risk ratio with SPEI-value is shown, supporting the notion that there is a relation between SPEI0 and the likelihood of a flood to occur. Nevertheless, as the editor suggests, the conclusions drawn by this analysis are carefully interpreted in the revised manuscript.

Furthermore, in order to address Reviewer #1 concern about the normalization of PRE7 and MAX7, we re-defined them in line with the definition of the Standardized Precipitation Index (SPI), by fitting a gamma distribution to the precipitation values that are used for the calculation of PRE7 and MAX7 (See also response to reviewer #1). Although, as expected, results do not change substantially, we follow editor's recommendation and we carefully interpret and explain the statistical significance that is found between floods and no-floods, and in the revised paper, we now discuss the possible explanations and conclusions of differences between PRE7 and MAX7.

**Reviewer #1:**

I reviewed the revised manuscript „Influence of antecedent conditions on flood risk in sub-Saharan Africa". The authors tried to extent their statistical analysis and compare the antecedent short- and seasonal scale conditions of reported floods by MunicRe with the conditions in years, where no flood occurred. Simple statistical tests are conducted in order to test the significance of precipitation anomalies at different temporal scales.
I appreciate the effort and I believe, that the manuscript has improved from a methodological point of view. However, I am unfortunately not convinced that the findings are yet sufficient for publication. The major conclusion, that floods are triggered by a combination of the catchment state (as represented by SPEI) and high precipitation during the build-up period is rather trivial and has been investigated in various studies, mostly by means of more complex methods. The differences reported in the manuscript are mostly not very clear and often not statistically significant. Further the shortcomings of the datasets and methods impede the interpretation of the results.

Thus, as much as I regret it, I cannot recommend publication at the current state. I would like to encourage the authors to further advance their methods (particularly to use more complex inferential methods, which are suitable for the data) and re-submit a new-version, if more robust results could be achieved.

We thank Reviewer #1 for his/her comments and we are happy that he/she finds that the manuscript has improved compared to the latest version. The reviewer has point, and in the revised version, we have further improved our statistical analysis, taking into account his/her comments.

As for the issue of existing research on the same topic, indeed, various studies have examined the influence of high precipitation during the flood build-up period in the past. However, to our knowledge, they do not explicitly distinguish between the weather- and the seasonal-scale flood build up period, which is particularly relevant for the alert function under data-sparse conditions in Africa. Moreover, they either conduct the analysis based on modelled floods, or they examine only very few flood events. Hence, by examining a relatively big number –compared to the previous studies- of real flood events, we have demonstrated that long-term antecedent conditions should not be a priori neglected as in several past events, they have played a role in flood mitigation.

Hence, we believe that the conclusions drawn in our revised version are both relevant and novel, as the paper integrates short- and long-term flood antecedent conditions based on a relatively large group of reported flood events. However, perhaps this was not clear in our discussion section, and we have further clarified the novelties of our methods as compared to existing literature

1)      Introduction, climatic and hydrological setting:

30 The study covers a large target region with various different climatic settings. No information is provided on climate-variability, large scale climatic circulation modes and on typical flood generation processes in different parts of the region. The Köppen classificastion (Fig.4) is not sufficient to describe the climate of the region (especially the very basic version, which classifies the center of the continent as oceanic?). It might be a step into the right direction to analyse the spatial climatic variability (SPEI). Maybe one could then find different flood types in different regions, which are characterized by different SPEI-flood relationships.

Although we have tried different climatological classifications, using also the latest versions of Köppen climatology, we did not manage to produce any statistically significant differences between the climate areas for the SPEI-flood relationship. By using the basic version of Köppen that includes only 3 climatological areas, we aimed to include as many floods as possible in each area, in order to increase our sample size. However, even in such a way, we did not find any statistically robust results. Identifying different flood types in different regions could be a step forward, but we believe that flood generation depends highly on local characteristics of both climatic and non-climatic nature. With the limited number of floods and reported characteristics we cannot derive a robust spatial classification of floods.

45

2)      Data Quality:

One major problem of the study remains the quality of the MunichRe data set. The
50 comparison of observed and non-observed floods assumes, that the data set is a) somehow complete and b) that the climate-flood link is stationary. However, as the authors admit, there is a strong trend in the data, which points to increasing settlements in flood-prone areas or an increase of flood reports. Those problems might blur important statistical relationship.

In their recent publication ("Should seasonal rainfall forecasts be used for flood
55 preparedness?") the authors show (based on modelling results), that the precipitation-flood link varies over the target region and is highly dependent on the climatic conditions. In the presented manuscript, all floods are pooled, which also might blur clear results. I fear, that the

quality MunichRe data set alone might be too poor and the number of reported floods too low to draw statistically significant conclusions and to derive interesting/new results.

We understand that Reviewer #1 is concerned about the quality of a reported database such as Munich Re. Throughout the manuscript, we have explicitly mentioned its weaknesses and how this might blur the statistical conclusions drawn, emphasizing that the analysis has been conducted, naming as 'floods' only the ones reported. To our knowledge, Munich Re is not only the most detailed dataset, it is also the most complete one. A recent thesis by Vries (2017) supports this claim, by showing that Munich Re presents the highest number of flood events in Tanzania, Malaysia and Ireland and it also includes the events reported in other databases such as FloodList, Flood Obseratory and EM_DAT. As we stated in the revised manuscript, we expect that by including the reported damaging flood events, this research will also be useful to the humanitarian sector.

Moreover, as Reviewer #1 mentions, our recent publication "Should seasonal rainfall forecasts be used for flood preparedness?" was based on modelling results and explored the usability of precipitation forecasts at seasonal time scales. Trying to validate/compare it with the real world, in this research we use reported floods, whose number is obviously much lower, and include explicitly the short-range forecasting time scale, which gives a clearly better description of the precipitation-flood relationship. To increase the statistical sample the events are pooled. Acknowledging the difficulties and the uncertainties of this, we believe that the novelty of this research, which examines the 'real world' instead of the modelling outputs, is of value.

In the text, we have checked all our statements on the accuracy of the datasets in detail, so as to make sure that they reflect all these issues.

3)      Statistical Methods:
There are quite a few methodological problems.

a)      The assumptions of statistical tests are often violated by data sets. E.g. event-precipitation (Pre7 and Max7) have been z-normalized, although short-term precipitation is certainly not normal-distributed. Likewise the significance of different mean SPEI values for flood and non-flood are based on a z-test and results are questionable.

We agree with Reviewer #1 that short-term precipitation is not normally distributed. In the analysis, we followed this simplification in order to assign a single numeric value to the short-term precipitation that can be compared across the different areas. In order to address Reviewer's #1 concern, we changed the way that PRE7 and MAX7 are defined; instead of a z-normalization, we defined them in line with the definition of the Standardized Precipitation Index (SPI), by fitting a gamma distribution to the precipitation values that are used for the calculation of PRE7 and MAX7. The cumulative probability calculated from the gamma distribution was transposed to the equivalent cumulative probability of the standard normal distribution. Hence, the resulting values are the standardization of total gamma-transformed accumulated precipitation values. Based on that, we applied a z-test to evaluate the difference from the medians of floods and no-floods.

b)      The 75%-quantiles in Fig. 6 clearly overlap, doesn't that actually indicate, that differences are not significant?

We use boxplots in order to show the (slightly) increased SPEI values of floods compared to the no-floods. We claim that the medians of the two groups are statistically different at the 5% level, since the comparison intervals (notches) do not overlap. Other definitions of significance may be used that come to different conclusions.

c)      Also the comparison of "flood probabilities" (Fig. 7) is not very robust, since it includes

classes with different numbers of cases. Thus, single floods with anomalous pre-conditions can change the entire plot (and thus the interpretation of results). E.g. the sudden drop of SPEI0 is certainly a statistical artefact. Again I would rather use boxplots or similar methods, which include the range of values and some measure of significance.

Indeed, the comparison of 'flood probabilities', which in the revised version is mentioned as 'risk ratio', includes different number of cases. For each case of Figures 1, 2, and 3 (see the end of this document), which correspond to Figures 7, 8 and 9 of the original manuscript, we have created a graph that shows the confidence intervals (Figures 4-19), as calculated by Moris and Gardner (1988), the results of the risk ratio are statistically significant in case the confidence interval does not include values below 1 (in our analysis).

Regarding the sudden drop of SPEI0, this probably happens due to the absence of flood events with SPEI0 larger than 3. In the new versions of the plots we don't show events for SPEI>2.5 due to the poor population of events in these extreme classes, which results in not statistically significant results

d)    Further I wonder, why flood-likelihood-ratio is always >1, even if SPEI is clearly negative. This might indicate either a data problem or a problem of the method.

In Figure 7, the ratio is either 1 or a bit higher than 1, when SPEI is negative. This is expected, as it compares (mutually exclusive) samples of floods and no-floods exceeding an SPEI-threshold are counted. Therefore, for SPEI = -3, the ratio is 1 and it slightly increases when SPEI -1, showing that there are proportionally slightly more floods than no floods exceeding at higher values of SPEI.

On the other hand, in Figure 8 and 9, the ratio is higher than 1 even in negative SPEI thresholds as this threshold is combined with positive seasonal SPEI and PRE7 thresholds, pointing at the importance of precipitation as a flood-generating process.

**Reviewer #2:**

Dear authors,

thanks for all your replies to my comments to your original manuscript (reviewer #2). Mostly thanks for accepting the advice of using box-plots in some of the figures.

In your comment you declare you adjusted everywhere the use of "lead time" and replaced it with antecedent conditions. I am not sure whether you missed to do it in Figure 3.

We thank Reviewer #2 of his/her substantial contribution on the improvement of this paper.

**Additional References**

Daly (1998) Confidence limits made easy: interval estimation using a substitution method. American Journal of Epidemiology 147: 783-790.

Hoeppe, P.: Trends in weather related disasters - Consequences for insurers and society, Weather Clim. Extrem., 11, 70–79, doi:10.1016/j.wace.2015.10.002, 2016.

Jongman, B., Hochrainer-Stigler, S., Feyen, L., Aerts, J. C. J. H., Mechler, R., Botzen, W. J. W., Bouwer, L. M., Pflug, G., Rojas, R. and Ward, P. J.: Increasing stress on disaster-risk finance due to large floods, Nat. Clim. Chang., 4(4), 264–268, doi:10.1038/nclimate2124, 2014a.

5      Katz, K. A.: The (relative) risks of using odds ratios., Arch. Dermatol., 142(6), 761–4, doi:10.1001/archderm.142.6.761, 2006.

Morris and Gardner (1988) Calculating confidence intervals for relative risks (odd rations) and standardized ratios and rated – Statistics in Medicine

Shrier, I. and Steele, R.: Understanding the relationship between risks and odds ratios., Clin. J. Sport Med., 16(2), 107–10, doi:10.1097/00042752-200603000-00004, 2006.

Vries, M. (2017). Het detecteren van overstromingen. BSc thesis, VU Amsterdam, Amsterdam

Zhang J, Yu KF. What's the Relative Risk? A Method of Correcting the Odds Ratio in Cohort Studies of Common Outcomes. *JAMA*. 1998; 280(19):1690–1691. doi:10.1001/jama.280.19.1690

**New Figures**

[Figure]

**Figure 1 Risk ratio between flood and no-floods likelihood as function of SPEI exceedance values. The circles are used**
25     **when the ratio is statistically significant and the triangles when it is not. (To replace figure 7 of the original**
**manuscript)**

[Figure]

**Figure 2 Risk ratio between floods and no-floods for given SPEI0 thresholds conditional to certain seasonal SPEI values. The circles are used when the ratio is statistically significant and the triangles when it is not. (To replace figure 8 of the original manuscript)**

[Figure]

**Figure 3 Risk ratio between floods and no-floods given PRE7 thresholds conditional to certain seasonal SPEI values. The circles are used when the ratio is statistically significant and the triangles when it is not. (To replace figure 9 of the original manuscript)**

**Figures for supplementary material**

**Confidence intervals for lines of Figure 1 (Figure 7 of the original manuscript)**

[Figure]

**Figure S.1 Confidence intervals for SPEI0 thresholds**

[Figure]

**Figure S.2 Confidence intervals for SPEI1 thresholds**

[Figure]

**Figure S.3 Confidence intervals for SPEI3 thresholds**

[Figure]

**Figure S.4 Confidence intervals for SPEI6 thresholds**

5     **Confidence intervals for lines of Figure 2 (Figure 8 of the original manuscript)**

[Figure]

**Figure S.5 Confidence intervals for SPEI1>0 and SPEI0 thresholds**

[Figure]

**Figure S.6 Confidence intervals for SPEI1>1 and SPEI0 thresholds**

[Figure]

**Figure S.7 Confidence intervals for SPEI3>0 and SPEI0 thresholds**

[Figure]

**Figure S.8 Confidence intervals for SPEI3>1 and SPEI0 thresholds**

[Figure]

**Figure S.9 Confidence intervals for SPEI6>0 and SPEI0 thresholds**

[Figure]

**Figure S.10 Confidence intervals for SPEI6>1 and SPEI0 thresholds**

**Confidence intervals for lines of Figure 3 (Figure 9 of the original manuscript)**

[Figure]

**Figure S.11 Confidence intervals for SPEI1>0 and PRE7 thresholds**

[Figure]

**Figure S.12 Confidence intervals for SPEI1>1 and PRE7 thresholds**

[Figure]

**Figure S.13 Confidence intervals for SPEI3>0 and PRE7 thresholds**

[Figure]

5    **Figure S.14 Confidence intervals for SPEI3>1 and PRE7 thresholds**

[Figure]

**Figure S.15 Confidence intervals for SPEI6>0 and PRE7 thresholds**

[Figure]

**Figure S.16 Confidence intervals for SPEI6>1 and PRE7 thresholds**

[revised manuscript text omitted]

precipitation (PRE7), comparable) of similar magnitude compared to the maximum observed 7-day precipitation of the flood onset months during the same month in the no-flood years. This indicates that although PRE7 was high, it is not able to fully justify the flood generation by itself, leading us to hypothesize that there should be other factors or explanations, other than intense rainfall, that have led to the flood event. These factors can be subject to either poor Alternatively, inaccuracies in the data used (i.e. reanalysis datasets, disaster database) or to the conditions that preceded the can also be (partly) of influence for not finding a strong relation between PRE7 event and flood events.

Despite the absence of high quality daily precipitation datasets in Africa (Lorenz and Kunstmann, 2012; Rogers and Tsirkunov, 2013; Zhang et al. 2013), precipitation reanalysis data offers valuable information over poorly monitored regions such as sub-Saharan Africa (Zhan et al. 2016). However, due to the lack of valuable ground-based precipitation records, especially in developing countries, the reliability of precipitation extremes in reanalysis datasets over land varies in location and time period and it can be very sensitive to reanalysis product and resolution choice (Herold et al. 2017). Particularly, the daily precipitation values on a coarse grid are largely uncertain as they do not capture local scale convective events, which are often responsible for high-intensity precipitation and could significantly affect our weather-scale results.

The rationale to perform the analysis over a large area around the reported flood coordinates is to deal with the uncertainty in the presented location of the reported flood and to capture the impact of the rainfall in neighbouring areas, including some upstream, which may have contributed to the flood generation mechanisms. Due to insufficient information in the disaster dataset, it is difficult to determine This simplified approach was necessary because we did not have the exact delineation of the upstream area. So, we followed this simplified approach. The real world is much more complicated, as the response of hydrological systems to precipitation varies considerably depending on time and place (Eltahir and Yeh, 1999). Further studies should give this serious consideration, carrying out analyses on local spatial scales and using hydrological models to estimate the travel and the concentration time of the upstream rainfall to each flood location.

Finally, in order to gain insights into the uncertainty of the flood onset date, we compared the maximum 7-day precipitation (MAX7) during the onset month of each flood, with a) PRE7 and b) the maximum 7-day precipitation. The median of no-flood events. In both cases, MAX7 was found to be significantly higher. This shows indicates that the 7 days prior to the reported onset date (PRE7) did do not always exhibit the highest precipitation during the flood month, as one might have expected. This means that either the flood reported date was not accurate or that the MAX7 worked complementary to PRE7 leading to the flood generation. (i.e. flooding was already triggered before the maximum 7-day precipitation had taken place). Again, focusing on a local scale, getting accurate information on the onset date, precipitation, discharges, etc. would be an important addition in future research.

*Role of seasonal-scale conditions*

Our results showed that the most reported floods were preceded by relatively wet seasonal conditions, as all the their SPEIs were greater than 0 (SPEI1-70%, SPEI3-65%, SPEI6-57%). Comparing the seasonal SPEI value of floods F events to that of no-floods NF events, we see that the median of the first is significantly higher than that of the second latter across the different seasonal timescales, (SPEI1 to SPEI6), indicating there were several cases that – in general - SPEI could have served as an early warning indicator, in case it had been monitored or forecasted. However, the median SPEI of floods goes towards climatological conditions for longer accumulation periods. This should be considered together with the decreasing forecast skill over the lead time (Molteni et al., 2011) in order to identify whether and at which point SPEI could be used as a flood warning indicator.

In a simple quantification of the flooding probabilities likelihood, we found used for the relative odds of floods first time in a flood risk research the risk ratio (RR), which is widely used in medical and no-floods epidemiology studies, comparing the

likelihood of F events to NF events under various SPEI thresholds. When using a threshold of 1.5 for SPEI1 and SPEI3, we found that it is arounda RR of 2.5 times more likely for a flood to occur., indicating an increased probability to encounter an F event. Although this number is not high, and the confidence intervals are quite wide, it is still first evidence that seasonal parameters could be used in flood warning systems. Using the samea threshold of 2 for SPEI0, which refers to the conditions during the flood onset month, we found that it wasthe RR becomes 6.5 times more likely a flood to have occurred. This shows that SPEI0 has captured in several cases the unusually wet conditions during the flood and that it could be used as a flood monitoring tool.

Finally, by bringing together the short- and the long-term conditions, we saw that the conditions during different time scales cancould possibly be used complementary to each other for flood warningswarning. Using thresholds for both seasonal SPEIs and SPEI0, the likelihood of having a floodan F event compared to a no floodan NF event is considerably increased compared to the same likelihood when taking into account only weather or seasonal scale conditions. For instance, when SPEI0 is above 2 and SPEI1, SPEI3 and SPEI6 are above 1, itthe RR becomes around 10, 12 and 14 times more likely to have a flood compared to a no flood. Nevertheless, SPEI0 refers to the entire month, when the flood was reported and not to the conditions that preceded its generation. Therefore, an early warning early action system could monitor rainfall and temperature observations, getting ready when the previous three months have had a high SPEI, and taking further action if the upcoming month is forecasted to also have a high SPEI.

On the other hand, when connecting PRE7 with seasonal SPEIs, the relative odds ratioRR did not exhibit so high values as before. However, the resultingthere is still considerably increased probabilities (around 4 and 5 times more likelyprobability of having an F compared to have a floodan 
[revised manuscript text omitted]

---

## Author Response (AR3)

***List of changes compared to the previous manuscript***

- Correction of typos
- Correction of references
5   - Small changes in the text
- Replace figure 4 (Köppen climate classification) with the figure that was used in the first submission of the manuscript, which shows the flood events per country and the number of fatalities they caused.
- Changes in figures 7, 8 and 9. The figures with the confidence intervals, which were in
10   the supplementary material in the previous revision, they have been incorporated into the new figures. Moreover, the entire range of the SPEI values is shown.
- Supplementary material has been removed.

30

35

40

45

50

55

**Response to reviewers and to the editor**

**Title:** The influence of antecedent conditions on flood risk in sub-Saharan Africa

**Authors:** Konstantinos Bischiniotis, Bart van den Hurk, Brenden Jongman, Erin Coughlan de Perez, Ted Veldkamp, Hans de Moel, Jeroen Aerts

**General response**

We thank the reviewer and the editor for their continued interest in our paper, and the comments they provided for the revision of our manuscript. In response to their comments, we have further revised our manuscript, and we have made changes as outlined in this document. In the following sections, we respond to their remarks and suggestions. Our responses are colored in blue. We think that with these revisions, the readability of paper has much improved.

**Editor:**

Dear authors,

as you have seen, the reviewer #1 is still not convinced that your analyses are statistically sound. However, given that the other reviewer is satisfied with the manuscript, and based on my own reading, I decide on minor revisions.

We thank the editor for his continued interest in our paper, and his comments and suggestions to further improve the manuscript. In the revised document, we have taken into account all the suggestions by the editor. We also addressed the comments by reviewer #1, which are described in more detail below.

When revising, please take into account the comments of reviewer #1. I strongly encourage you to consider his comments on the representation of uncertainty ("... Concerning the illustration of the results: The uncertainty of the results should be well communicated. Thus confidence intervals should always be shown in the manuscript (not only in the appendix). I would suggest to show (light colored) polygons behind each line in the original figures and delete all the confindence plots of the supplementary material. ... Further, the entire range (including the rather strange behavior for SPI0=3) should be shown in order to communicate the high uncertainty of the results ..."). Of course, this hint should not degrade the other comments of reviewer #1.

We included the confidence intervals in the original manuscript and we have removed the supplementary material.

Please check the whole manuscript carefully! It contains quite a few typos. Further, the papers of Golnaraghi (2010), Braman et al. (2013) and the 3 papers of Coughlan De Perez et al. are not listed in the references, and the reference to Rogers and Tsirkunov (2010) is wrong. I stumbled across these errors by chance, so there might be a high probability that there are other errors in
the manuscript.

We went through the entire manuscript and the reference list, and corrected the typos and mistakes.

**Reviewer #1:**

I once again reviewed the revised version of the manuscript of Bischiniotis at al. Personally I am unfortunately still not very confident with the quality of the data and particularly with the robustness of the applied methods. However I acknowledge the fact the reviews of the manuscript diverge a lot (the other expert seems to be satisfied) and thus would like to hand over the decision to the editor.

We thank the reviewer for his/her time taken to review again our manuscript. Despite the reviewer is still not fully satisfied, we further addressed the comments that are addressed in the review, such as the representation of uncertainty and the Köppen classification. Regarding the quality of the data, we understand the concerns that he/she expresses. However, we believe, as we have explicitly mentioned in our manuscript by also citing other studies, that for this type of research the datasets used are the best available. We have now also included further representations of uncertainty to several figures. Below, we respond to each of the individual comments;

Please find my major concerns below:

1) Climatic settings:
The authors still refer to the very basic Köppen climate classification, which assigns some parts of Central Africa to "oceanic climates". I think the authors mean "summer moist subtropical climates with a pronounced seasonality". In general I do not see any benefit from the Köppen climate map and a source is still missing.

We agree with the reviewer that referring to some part of central Africa as 'oceanic climates' is not correct. We have found in literature that this should be referred as 'mild temperate' climate. However, we also agree that the Köppen classification does not give any additional value to the analysis. During the revision process we tried to find correlations and differences between the different climate areas. Since we did not manage to find any statistical significant results, we decided to remove the Köppen map and replace it with the figure that we used in the first revision, which shows the flood events in each country and the number of fatalities they caused.

2) Statistical Methods:
As stated in the former review, I am not convinced that the methods applied are robust and I feel that statistical assumptions of parametric methods are often violated.
The normalization of PRE7 is now done following the SPI methodology. Therefore 31 data points (including one flood and 30 non-flood cases) are fitted by means of a gamma distribution. Since different types of cases are combined (f, nf cases), I do not think that the application of a parametric method is feasible. In general the applicability of the distribution should be tested.

The same applies to the generation of confidence intervals of the RR: Here the authors state, that the logarithm of the sample is normally distributed. This is not trivial to me and has to be shown or justified.

For the normalization, we followed the methodologies followed in SPI and RR that have been used in literature. Flood and No-flood events are considered to be related to antecedent conditions that are shaped by (multiday) precipitation events. These events emerge from a long univariate meteorological time series that can be analyzed with the parametric methods, as is demonstrated in many studies using SPIs. Therefore, we think that the methodology can be applied.

Concerning the illustration of the results: The uncertainty of the results should be well communicated. Thus confidence intervals should always be shown in the manuscript (not only in the appendix). I would suggest to show (light colored) polygons behind each line in the original figures and delete all the confidence plots of the supplementary material. Further, the entire range (including the rather strange behaviour for SPI0=3) should be shown in order to communicate the high uncertainty of the results.

We now included the confidence intervals in the original manuscript with light color polygons as suggested and we have removed the supplementary material.

3) Conclusions
The conclusions and the discussion are almost congruent. I think one can delete the latter.

We have revised again the conclusions and we believe that they reflect our main findings.

[revised manuscript text omitted]